

# Multiyear chemical composition of the fine aerosol fraction in Athens, Greece, with emphasis on winter-time residential heating

Christina Theodosi[1,2], Maria Tsagkaraki[1], Pavlos Zarmpas[1], Eleni Liakakou[2], Georgios Grivas[2], Despina Paraskevopoulou[2], Maria Lianou[2], Evangelos Gerasopoulos[2], Nikolaos Mihalopoulos[1,2]

[1]Environmental Chemical Processes Laboratory (ECPL), University of Crete, Heraklion, Crete, 71003, Greece
[2]Institute for Environmental Research and Sustainable Development (IERSD), National Observatory of Athens, P. Penteli, Athens, 15236, Greece

*Correspondence to*: N. Mihalopoulos (mihalo@uoc.gr) and C. Theodosi (c_theodosi@chemistry.uoc.gr)

**Abstract.** In an attempt to take effective action towards mitigating pollution episodes in the Greater Athens Area (GAA),

precise knowledge of $PM_{2.5}$ composition and their sources is a prerequisite. Thus, a two year chemical composition data set from aerosol samples collected in an urban-background site of central Athens, from December 2013 till March 2016, has been obtained and Positive Matrix Factorization (PMF) was applied in order to identify and apportion fine aerosols to their sources. A total of 850 aerosol samples, were collected on a 12 to 24h basis and analyzed for major ions, trace elements, organic and elemental carbon, allowing us to further assess the impact of residential heating as a source of air pollution over

the GAA.

The ionic and carbonaceous components were found to constitute the major fraction of the $PM_{2.5}$ aerosol mass. The annual contribution of the Ion Mass (IM), Particulate Organic Mass (POM), dust, Elemental Carbon (EC) and Sea Salt (SS) were calculated at 31%, 34%, 18%, 8% and 3%, respectively. However, carbonaceous aerosols (POM + EC) and IM exhibited considerable seasonal variation. In winter, IM was estimated down to 23%, with POM + EC being the dominant component

accounting for 48% of the $PM_{2.5}$ mass, while in summer IM was the dominant component (42%), followed by carbonaceous aerosols 37%.

Results from samples collected on a 12h basis (day and night) during the 3 intensive winter campaigns indicated the impact of heating on the levels of a series of compounds. Indeed $PM_{2.5}$, EC, POM, $NO_3^-$, $C_2O_4^{2-}$, $nssK^+$ and selected trace metals including Cd and Pb were increased by almost a factor of 4 during night compared to day, highlighting the importance of

heating on air quality of the GAA. Furthermore, in order to better characterize winter-time aerosol sources in the city centre of Athens and quantify the input of biomass burning as a source to winter night-time $PM_{2.5}$ concentrations, source apportionment was performed. The data can be interpreted on the basis of six sources namely biomass burning (32%), vehicular emissions (19%), heavy oil combustion (7%), regional secondary (20%), marine aerosol (9%) and dust particles (8%). With specific emphasis on night to day contrasts their contributions shifted from 19, 19, 8, 30, 11 and 9% of the $PM_{2.5}$

mass during day to 39, 19, 6, 14, 7 and 6% during night, underlining the significance of biomass burning as the main contributor to fine particle levels during night-time.



## 1 Introduction

The scientific interest on aerosols has widely increased during the last decades due to their impact on air quality, human health and climate change (e.g. Seinfeld and Pandis, 1998). Legislation regarding atmospheric particulate matter is gradually becoming more stringent, as a result of the frequent, high aerosol episodes encountered on regional or even continental scales, associated with synoptic and mesoscale meteorological conditions (Querol et al., 2009). Significant efforts are hence targeted towards improving air quality through measures for emissions reduction (Daskalakis et al., 2016).

Particles with diameter of 2.5 μm or less are of particular interest due to the fact that they contribute significantly to detrimental health effects (Dockery and Pope, 1994; Ostro et al., 2006). This is because they penetrate more efficiently the cell membranes (Salma et al., 2002; Li et al., 2003; Bell et al., 2009) and act as carriers of toxicants and mutagenic components (Beddows et al., 2004). Recent epidemiologic studies have highlighted the risk of exposure to enhanced levels of carbonaceous aerosols, revealing notable associations with cardiovascular mortality and morbidity (Ostro et al., 2010; Lipsett et al., 2011; Krall et al., 2013). Trace metals are also connected with chronic and acute health problems due to their toxicity (Chapman et al. 1997; Wichmann and Peters 2000; Pope et al. 2002; Stiebet et al. 2002; Theodosi et al. 2011). As a consequence, effective action to mitigate pollution episodes depends much upon the deep understanding of the relevant emission sources, which in turn calls for specialized measurements and advanced statistical approaches of source apportionment.

In Greece, air quality has improved since the advent of the global economic recession in 2008, due to the abrupt cut down of anthropogenic activities such as vehicles use and industrial production (Vrekoussis et al. 2013; Paraskevopoulou et al., 2014, Gratsea et al., 2017). However, since winter 2011-12 the massive use of wood as fuel for residential heating changed this winter-time trend (e.g. Gratsea et al. 2017). Burning wood and coal in residential stoves (and fireplaces) is an important source of directly emitted fine particulate matter ($PM_{2.5}$), EC and polycyclic aromatic hydrocarbons (PAH), with great impact on air quality (EEA, 2013; 2014). Paraskevopoulou et al. (2015) have shown that at a suburban site (Penteli) in the Greater Athens Area (GAA), the contribution of Particulate Organic Matter (POM) to the additional local aerosol mass, increased by 30% between winter 2012. Fourtziou et al. (2017) have then reported on several tracers of wood burning monitored during winter 2013-2014, linking them with the presence of severe smog events due to wood combustion for residential heating.

Informed decision making towards improving air quality, demands precise knowledge of PM chemical speciation and sources attribution. Based on the effect of the finer particles on health and their relevance to urban sources, in contrast to natural aerosols, it is even more important to focus such analyses on fine fractions of aerosol. In this study, $PM_{2.5}$ was chemically characterized for inorganic species, such as trace elements and water soluble ions, as well as for carbonaceous components, such as organic and elemental carbon. To our knowledge, this is the first time that such a long-term, uninterrupted estimation of the chemical composition of $PM_{2.5}$, a chemical mass closure exercise and source identification of particulate matter, took place in parallel, at an environment in the Southern Europe offering challenging conditions in terms



of pollution contributors and timing (crisis years). Given the intensive use of wood as fuel for residential heating since winter 2012 in Athens, the current work emphasizes on winter periods. During three consecutive winters (2013-2014 to 2015-2016), aerosol sampling was intensified, from routine 24h time resolution to 12h resolution, in order to separate day-time from night-time levels and highlight the impact of this "new aerosol source" on urban Athens' air quality.

## 2 Experimental

### 2.1 Sampling site and climatic aspects of the region

Aerosol sampling was conducted on a daily basis at the central premises of the National Observatory of Athens, downtown Athens, (Thissio, 38° 0.00′ N, 23° 43.48′ E, 110 m a.s.l.). This urban background site is not directly impacted by local human activities as it is surrounded mostly by a pedestrian zone and moderately-populated neighbourhoods, with light vehicle traffic and other immediate local sources. Therefore, it is considered representative for the exposure of the majority of population of the Greek capital. The major sources of air pollution affecting the site are expected to be mainly residential heating and vehicular emissions from the city center and, while the impact from long-range transport of continental air masses from Central and Eastern Europe should not be neglected (Gerasopoulos et al., 2011; Perrone et al., 2013; Bougiatioti et al., 2013).

The GAA is a region characterized by enhanced concentrations of particulate matter originating from both locally emitted sources (Kanakidou et al., 2011) and long range transport (Koulouri et al., 2008; Theodosi et al., 2011; Gerasopoulos et al., 2011; Remoundaki et al., 2013). On a yearly basis, the Northern sector is the dominant for arriving air masses (originating from Central and Eastern Europe), accounting for almost two thirds of the time (Theodosi et al., 2011; Gerasopoulos et al., 2011). The Southern sector is linked to Sahara dust transport (S/SW winds; occurrence up to 25%) and dominates in frequency during the transition seasons. The ventilation of the basin is poor during the prevalence of local circulation systems, such as sea/land-breezes along the major NE-SW geographical axis of the basin. A detailed description of Athens prevailing meteorology, morphology and internal transport patterns in the urban complex are provided by Kanakidou et al. (2011) and references therein.

### 2.2 Sampling and chemical analyses

$PM_{2.5}$ aerosol was collected on Quartz fiber filters (Flex Tissuquartz, 2500QAT-UP 47mm, Pall) with a Dichotomous Partisol Sampler 2025 (Rupprecht & Patashnick, 16.7 L min$^{-1}$), on a daily basis during a period of more than two years (December 2013 - March 2016), resulting to the collection of 848 samples. During the three occurring winter periods (from December to February), two-month intensive campaigns (sampling frequency of 12h; n=447) were conducted, in an attempt to study in depth the characteristics of emissions from heating activities. The $PM_{2.5}$ aerosol mass was gravimetrically determined (samples were weighed before and after sampling, under controlled conditions) using a microbalance as described by Fourtziou et al. (2017) and Paraskevopoulou et al. (2014), and filters were stored until the chemical analysis.



Filter blanks and blank field samples were also prepared and analyzed. All $PM_{2.5}$ samples were analyzed for organic (OC) and elemental carbon (EC), water soluble ions ($Cl^-$, $Br^-$, $NO_3^-$, $HPO_4^{2-}$, $SO_4^{2-}$, $C_2O_4^{2-}$, $Na^+$, $NH_4^+$, $K^+$, $Mg^{2+}$ and $Ca^{2+}$), crustal origin (Al, Fe, Ca) and trace elements (Zn, Pb, Cu, Ni, V, Cr, Mn, Cd, As). The chemical speciation data were utilized to perform a chemical mass closure exercise and chemometric receptor modelling for source apportionment.

Quartz filters were analyzed for OC and EC, with the Thermal-Optical Transmission (TOT) technique (Birch and Cary, 1996), using a Sunset Laboratory OC/EC Analyzer, as described in detail by Theodosi et al. (2010a) and Paraskevopoulou et al. (2014) and applying the EUSAAR-2 protocol (Cavalli et al., 2010). In short, during the first phase (OC phase) the sample (1 or 1. 5 $cm^2$) is kept in helium atmosphere and heated in four steps; 200, 300, 450 and 650 °C, while during the second phase, four consecutive temperature steps (500, 550, 700 and 850 °C in $He/O_2$ atmosphere) are applied. The detection limit
of the analysis was 0.26 and 0.05 µg C $cm^{-2}$ for OC and EC, respectively, and the reported results were blank corrected.

Filter parts were analyzed by ion chromatography (IC) for the determination of the main ionic species concentrations ($Cl^-$, $Br^-$, $NO_3^-$, $HPO_4^{2-}$, $SO_4^{2-}$, $C_2O_4^{2-}$, $Na^+$, $NH_4^+$, $K^+$, $Mg^{2+}$ and $Ca^{2+}$), as described by Paraskevopoulou et al. (2014). First, they were extracted in ultrasonic bath and then filtered using syringe filters (0.45µm pore size), in order to remove any insoluble species.

An acid microwave digestion procedure, followed by Inductively Coupled Plasma Optical Emission Spectrometry (ICP-OES, Thermo Electron ICAP 6000 Series), was applied for the determination of major and trace metal concentrations during this long-term sampling period (n=848) following the procedure described in details by Theodosi et al. (2010b). All 12 elements, Al, Ca, Mn, Fe, V, Cr, Ni, Cu, Zn, Cd, As and Pb, were also determined by Inductively Coupled Plasma Mass Spectrometry (ICP-MS, Perkin Elmer, NexION 300X) for all our winter (December-February) and summer (June-August)
samples (n=592). Results reported hereafter for Al, Ca, Mn, Fe, V, Cr, Ni, Cu, Zn correspond to ICP-OES analysis, while Cd, As and Pb to ICP-MS. Yttrium (Y) and Indium (In) were added as an internal standard to the samples before ICP-OES and ICP-MS analysis, respectively. All reported concentrations were corrected for blanks.

Hourly meteorological data on horizontal wind velocity and prevailing wind direction (divided in 16 sectors) from NOA's station at Thissio were additionally retrieved.

**2.3 Source apportionment**

In order to identify major winter-time sources of $PM_{2.5}$ and their day-night contrasts, PMF receptor modelling was performed on 12-h winter chemical composition data. Days with availability of both day-time and night-time chemical speciation data were included in the PMF dataset.

In the PMF factor analytic model, speciated sample data are decomposed into matrices of factor contributions and factor
profiles. The matrix elements are obtained through the minimization of weighted decomposition residuals (object function - *Q*) in an iterative process. Uncertainties associated with analysis of individual species are used for weighting. In the present case, the multilinear engine (ME-2) program was used for solving the PMF problem in the setting of the EPA PMF5.0 software.





Uncertainties per species and sample were calculated (Reff et al., 2007) based on the error fraction of the measurement and the method detection limit for each component (MDL) according to [1]:

$$Unc. = \sqrt{(error\ fraction \ \times concentration)^2 + MDL^2} \qquad [1]$$

Values equal to 5/6th of the detection limit were assigned as uncertainty to samples below detection limits (BDL). Missing data points were substituted by the geometric mean of the respective species concentrations and given four times this value as uncertainty. $Mg^{2+}$ and As were excluded from the PMF analysis due to the large number of BDL samples, which resulted at signal-to-noise (S/N) ratios below 0.5. $Na^+$, $Ca^{2+}$ and Cd were included as down-weighted variables having their uncertainties tripled. $PM_{2.5}$ concentration was set as a total variable and had its uncertainty tripled so that it would not overly influence solutions. The PMF analysis was performed on the combined day/night dataset to obtain the source profiles and their contributions to $PM_{2.5}$ were calculated separately (Bernardoni et al., 2011; Canha et al., 2014).

Solutions involving 4-10 sources were examined and the ratios of actual (robust) to expected values of Q ($Q_R/Q_{EXP}$) were recorded. The selected solution, which was obtained for six factors, was physically interpretable, and the reduction of the $Q_R/Q_{EXP}$ for solutions with a greater number of factors was small, indicating that new factors were not introducing additional information (Brown et al., 2015). The physical validity of the solution was assessed in terms of within-source species ratios, meaningful day-night variability and correlations with external tracers (gaseous pollutant concentrations and aerosol metrics). The impact of small-medium scale atmospheric circulation has also been taken into account, by examining associations between source contributions and wind direction and velocity, through conditional bivariate probability (CBPF) function analysis. CBPF calculations and graphical interpolation in polar coordinates has been performed according to the methodology developed by Uria-Tellaitxe and Carslaw (2014).

The stability of the obtained solution against random sampling errors and rotational ambiguity was assessed using the BS-DISP procedure of the EPA PMF 5.0 (Paatero et al., 2014). One hundred bootstrap runs were performed ($R^2$>0.6 required for valid mapping) and the main identifying component from each of the six factors was displaced. Error intervals (EE) for displaced species were calculated as the ratio of the 95th- 5th percentile difference to the mean concentration of displaced species, from resamples of the BS-DISP process.

## 3 Results

### 3.1 PM2.5 levels

The daily $PM_{2.5}$ mass concentration at the urban background site of Thissio varied significantly from 0.3 to 144 $\mu g\ m^{-3}$, with higher concentrations occurring in the winter (27.4±8.7 $\mu g\ m^{-3}$) and lower in the summer (14.7±1.2 $\mu g\ m^{-3}$) (Figure 1).

The annual $PM_{2.5}$ concentration of the two complete years of the study period (2014 and 2015), based on daily values, were equal to 22.7±16.4 and 19.3±16.1 $\mu g\ m^{-3}$, respectively, both being lower than the annual $PM_{2.5}$ limit imposed by the EU



Ambient Air Quality Directive (2008/50/EC), which is set at 25 µg m⁻³. During the three winter periods studied about 35-43% of the values are above the target value, while on average 5% during the rest of the seasons.

The PM₂.₅ values reported here are in good agreement with those reported in other urban environment studies, for Athens (18-26 µg m⁻³, Theodosi et al., 2011; Mantas et al., 2014, Paraskevopoulou et al., 2015) or other European cities such as Spain, Italy, Czech Republic etc.  (20-30 µg m⁻³, Putaud et al., 2010).

### 3.2 PM₂.₅ chemical composition

### 3.2.1 Carbonaceous components

Figure 2a represents time series of the daily concentration levels of OC, which ranged from 0 to 49.5 µg m⁻³ (annual mean: 4.0±2.0 µg m⁻³; median 3.1 µg m⁻³; Table 1), and of EC, from 0 to 19.3 µg m⁻³ (annual mean: 1.5±1.0 µg m⁻³; median 1.1 µg m⁻³). At the regional scale, both species are emitted by common, primary sources, whereas OC is also associated to secondary aerosol production. Both carbonaceous components, exhibited a distinct seasonal variability, with lower concentrations during summer, in the order of 2.9±0.1 µg m⁻³ and 0.7±0.1 µg m⁻³ for OC and EC, respectively. Whilst, winter values were higher and equal to 6.3±2.9 µg m⁻³ and 2.8±1.2 µg m⁻³, respectively.

OC and EC concentrations fall within the range reported for background urban sites across Europe (Spain - Sanchez de la Campa et al., 2009 and Viana et al., 2007, OC=2-2.9 µg m⁻³ and EC=0.54-1.1 µg m⁻³; Germany, Prague, Amsterdam, Helsinki - Sillanpää et al., 2006, EC=0.7-1.69 µg m⁻³). Their levels are higher than those reported for a remote background site in Greece (Koulouri et al., 2008; OC=1.8±1.4 µg m⁻³ and EC=0.27±0.18 µg m⁻³) and a suburban site in Athens (Remoundaki et al., 2013; OC=2.43 µg m⁻³ and EC=0.99 µg m⁻³), while significantly lower than those previously reported for megacities such as Istanbul (Theodosi et al., 2010a; OC=6.6 µg m⁻³ and EC=2.9 µg m⁻³).

### 3.2.2 Ionic Composition

$SO_4^{2-}$ was the main ion contributor to the fine aerosol mass accounting for 16% (annual mean: 3.0±0.8 µg m⁻³; median 2.9 µg m⁻³), while $NH_4^+$ and $NO_3^-$ followed with contribution of 7% (mean: 1.4±0.7 µg m⁻³; median 1.11 µg m⁻³) and 2% (mean: 0.5±0.4 µg m⁻³; median 0.3 µg m⁻³), respectively. Figure 2b represents the daily variation of $SO_4^{2-}$ and $NO_3^-$, while Table 1 provides the annual levels of all the examined ions.

The daily concentrations of $SO_4^{2-}$ at the urban background site of Thissio vary from 0.5 to 12.5 µg m⁻³. These results are in excellent agreement with the value (3.1±0.8 µg m⁻³) reported by Paraskevopoulou et al. (2015) for the suburban station of Penteli in the GAA and also with other studies performed in Athens (4.0 µg m⁻³, Remoundaki et al., 2013; Mantas et al., 2014). For $NH_4^+$ and $NO_3^-$, the annual measured levels of $NO_3^-$ at Thissio are significantly higher from those reported for suburban and background locations in Greece, while for $NH_4^+$ they are comparable to literature data for Greece (07-1.5 µg m⁻³ for $NH_4^+$ and 0.1-1.1 µg m⁻³ for $NO_3^-$, Koulouri et al., 2008; Theodosi et al., 2011; Remoundaki et al., 2013, Mantas et al.,



2014; Paraskevopoulou et al., 2015), highlighting their respective role as local ($NO_3^-$) and regional ($NH_4^+$; see below) contributing sources, respectively.

Several other water soluble ions were also identified such as $Cl^-$, $Br^-$, $HPO_4^{2-}$, $C_2O_4^{2-}$, $Na^+$, $K^+$, $Mg^{2+}$ and $Ca^{2+}$ with an annual mean in the order of $272\pm197$ ng m$^{-3}$, $19.4\pm10.7$ ng m$^{-3}$, $54.3\pm37.4$ ng m$^{-3}$, $143\pm56$ ng m$^{-3}$, $307\pm196$ ng m$^{-3}$, $273\pm109$ μg m$^{-3}$, $64.0\pm31.6$ ng m$^{-3}$ and $160\pm127$ ng m$^{-3}$, respectively (Table 1). All the reported annual concentrations of the studied ions are in the same range as those reported for PM$_{2.5}$ in Athens over different sampling periods and stations (Theodosi et al., 2011; Pateraki et al., 2012; Remoundaki et al., 2013; Paraskevopoulou et al., 2015).

### 3.2.3 Trace metals

Table 1 summarizes the mean annual concentrations of elements and trace metals during the sampling period and Figures 2c-f represent the daily variation of several representative metals. The mean annual concentrations of natural elements of crustal origin such as Al, and elements of mixed origin, still with a significant crustal component, such as Fe and Ca, vary from 0.26 to 0.75 μg m$^{-3}$ (Table 1). The mean annual values for elements of anthropogenic origin (Mn, V, Cr, Cd, Ni, Cu, As, Pb) are generally very low, varying from 1.7 to 27.1 ng m$^{-3}$. On a monthly basis, the concentrations of toxic metals originating from human activities, such as As, Cd, and Ni, which are mainly confined in PM$_{2.5}$ fraction (Koulouri et al., 2008), they do not exceed few ng m$^{-3}$.

Compared to values reported in earlier studies for other locations in the GAA, it is concluded that during the last decade trace element concentrations have remained in the same order of magnitude, with only slight differentiation per case (Karanasiou et al., 2009; Theodosi et al., 2011; Pateraki et al. 2012; Remoundaki et al., 2013; Mantas et al., 2014; Paraskevopoulou et al., 2015). More specifically, as presented in Table 1, the majority of metals at Thissio are similar to those from a 5-year study at a suburban station in the GAA (Paraskevopoulou et al., 2015). As expected, the values are higher than those reported for several other rural background locations around Europe and Greece (Salvador et al., 2007; Koulouri et al., 2008; Viana et al., 2008; Pey et al., 2009; Alastuey et al., 2016).

### 3.3 Chemical Mass Closure

From the aerosol chemical components measured here: IM, dust, POM, EC and SS, the mass closure of the PM$_{2.5}$ aerosol samples can be undertaken. IM was calculated as the sum of the main anions and cations measured by ion chromatography, excluding the soluble $Ca^{2+}$ cation which is accounted in the dust fraction. Dust was estimated using Al (Ho et al., 2006) assuming an upper crust relative ratio of 7.1% (Wedepohl, 1995). POM was estimated by multiplying the measured OC with a conversion factor (CF), which corresponds to the ratio of organic mass to OC. CF is usually in the range from 1.4 for urban aerosol to 1.8 for remote aerosol. Turpin and Lim (2001) revisited these CFs and proposed values of $1.6\pm0.2$ and $2.1\pm0.2$ for urban and non-urban aerosol, respectively. Following that, a CF of 1.6 was applied in this study. SS originating species were calculated from the sum of the measured ions: $Na^+$, $Cl^-$, $Mg^{2+}$, ss$K^+$, ss$Ca^{2+}$ and ss$SO_4^{2-}$ (Sciare et al., 2005; Pio et al., 2007).



Carbonaceous aerosol constitutes the major component of the PM$_{2.5}$ mass, followed by sulphate and ammonium ions, dust and EC (Figure 3). More specifically on an annual basis, POM contributes 34% to the total PM$_{2.5}$ mass, while EC 8%. IM accounts also for a significant part of the PM$_{2.5}$ mass, equal to 31%, with SO$_4^{2-}$ (16%) and NH$_4^+$ (7%) being the dominant ions (Figure 4). The annual contribution of dust and SS is 18% and 3%, respectively. The corresponding chemical mass closure can explain about 95% of the measured, fine aerosol mass, leaving out a considerably low proportion of the unaccounted mass, which is usually water (Ohta and Okita, 1990). By comparing the aerosol mass (determined from the filter weighting, PM$_{2.5}$ mass) and the sum of individual chemical aerosol components a significant correlation is revealed, with a slope equal to 1.04 (r=0.88, n=780; not shown).

POM and IM present considerable seasonal variation. The results of the mass closure exercise for PM$_{2.5}$ on a seasonal basis are illustrated in Figure 4. In winter, IM is reduced (down to 23%), the dominant component being POM (38%), and the rest is shared by dust (18%), EC (10%) and SS (4%). In summer, IM is the dominant component (42%), followed by POM (32%), dust (24%), EC (5%) and SS (4%).

### 3.4 Temporal variability of winter mass and aerosol chemical composition: the role of residential heating

### 3.4.1 PM$_{2.5}$ mass

A significant increase in fine aerosol mass was observed in the GAA during winter (27.4 μg m$^{-3}$) compared to summer (14.7 μg m$^{-3}$), pointing towards an important PM source during winter (Figures 4a, b and 5a, b). In fact, during winter-time, residential heating using fossil fuel, wood and coal are important sources of directly emitted PM$_{2.5}$ (EEA, 2013; 2014). Moreover, the winter stable atmospheric conditions in conjunction with the seasonal decrease of the boundary-layer height (low winds, temperature inversion and solar radiation) could further favour the low dispersion of pollutants. Thus, elevated local emissions due to intense anthropogenic activities in the GAA, such as residential heating, can account for the observed increase in PM$_{2.5}$ levels in winter compared to summer.

This is more evident in the seasonal variation of Black Carbon (BC) and its wood burning fraction (BC$_{wb}$), obtained with the use of an aethalometer (AE33) during the period 2015-2016 (Figure 6). Both BC and BC$_{wb}$ are significantly higher during winter, reflecting the increasing impact of wood burning in PM levels in GAA (e.g. Gratsea et al., 2017; Fourtziou et al., 2017). High levels of PM mass during winter due to wood burning have also been observed in prior studies in the two largest urban metropolitan cities in Greece, Athens and Thessaloniki (Paraskevopoulou et al., 2014; 2015; Saffari et al., 2013; Fourtziou et al., 2017; Gratsea et al., 2017; Diapouli et al., 2017).

It is noteworthy that during winter, PM$_{2.5}$ concentrations during night-time (mean 32.9 μg m$^{-3}$) are almost twice as high (80% increase; Table 2) as during day-time (mean 19.1 μg m$^{-3}$), which constitutes additional evidence for the role of domestic heating in the night during winter (Figure 7a). Using the approach introduced by Fourtziou et al. (2017), i.e by selecting periods with wind speed lower than 3 m s$^{-1}$ and absence of precipitation, a series of days with smog events (hereafter called as SP; Smog Period; n=289) associated with increased air pollutants (NO, CO, BC) have been identified during the 3





examined winters. By further studying the PM$_{2.5}$ concentrations during these smog events, a 96% increase during night-time compared to day was observed (Table 2, statistically significant at 99.9% level; p<0.001).

### 3.4.2 Carbonaceous components

Both OC and EC present higher values during winter (Figures 5c-f). Primarily emitted OC and EC from residential heating,
in conjunction with strong low-altitude temperature inversions that trap pollutants near the surface, can explain their net seasonal trend (Kassomenos and Koletsis, 2005). Indeed OC and EC levels during winter-time are higher by 55 and 74%, respectively, compared to summer. In summer, POM (4.7 µg m$^{-3}$; 32% of PM$_{2.5}$) and EC contributions (0.7 µg m$^{-3}$; 5% of PM$_{2.5}$) are less pronounced than in winter POM (10.3 µg m$^{-3}$; 38% of PM$_{2.5}$) and EC (2.8 µg m$^{-3}$; 10% of PM$_{2.5}$), reflecting their common sources (such as domestic heating) and the importance of winter-time sources in the GAA. Note also the much
higher variability of both OC and EC during winter-time (Figures 5c-f) emphasizing the intensity and the sporadic nature of this source.

To highlight the importance of heating on carbonaceous levels, Figures 7b and c present their day and night-time variability in winter. The average OC and EC concentrations increased 3 and 2 times during night (Table 2). Similar tendency is observed during smog events (SP, Table 2). More specifically, the average OC and EC concentrations during the night for all
three winter campaigns are equal to 9.4 µg m$^{-3}$ and 3.8 µg m$^{-3}$ (12.6 µg m$^{-3}$ and 5.1 µg m$^{-3}$ for SP), respectively, with day values between 2.7 µg m$^{-3}$ and 1.6 µg m$^{-3}$ (3.4 µg m$^{-3}$ and 2.0 µg m$^{-3}$ for SP), respectively. Consequently, the contribution of POM to the total mass of PM$_{2.5}$ in winter is higher during the night (47±4%, median 45%) than in the morning (24±9%, median 21%), which constitutes additional evidence for the role of domestic heating (fuel, wood and biomass burning; Figure 7b, c) on fine aerosol levels. Similarly for EC, in winter a smaller but yet evident increase was also observed during
night (12±1%, median 12%) compared to day (8±3%, median 9%).

The significant correlation between OC and EC in the GAA during winter (slope=2.36; r=0.94; n=472), more enhanced during night-time (slope=2.49; r=0.96; n=226) compared to day-time (1.62; r=0.85; n=221), indicates that both carbonaceous components originate from the same sources. Interestingly, OC and EC anti-correlate in summer (3.21; r=-0.59; n=114). The higher OC to EC ratio during summer, as well as their negative correlation, could be explained by a larger photochemical
organic aerosol formation in the atmosphere, from low-volatility compounds produced by the oxidation of gas-phase anthropogenic and biogenic precursors. A significant correlation with PM$_{2.5}$ mass was also evident in winter period for both carbonaceous components (r=0.81, n=478), as opposed to summer (r=0.56 and 0.64).

### 3.4.3 Ionic Composition

SO$_4^{2-}$ concentration in the PM$_{2.5}$ fraction of aerosol increases gradually from spring to summer (Figure 2b). During the dry
season (spring and summer), the absence of precipitation, and the increased photochemistry lead to secondary aerosol formation and increased lifetime in the area (Mihalopoulos et al., 1997) and hence the appearance of higher concentrations. A secondary SO$_4^{2-}$ maximum is commonly recorded from November to March, coinciding with the seasonal decrease of the



boundary-layer height (low winds, temperature inversion and solar radiation). In winter, $SO_4^{2-}$ accounts for 8% of the $PM_{2.5}$ mass, while in summer for 26% (Figure 4a, b). During winter, $SO_4^{2-}$ did not show any significant day to night variability (about 11% increase during SP; Table 2), indicating that heating is not the major source of $SO_4^{2-}$. In addition, the maximum during summer time suggests that the majority of $SO_4^{2-}$ originates from long range transport and thus it can be considered as

an indicator of regional sources (Mihalopoulos et al., 1997; Theodosi et al., 2011).

The concentration of **$NH_4^+$** presents a less pronounced seasonal trend, with a similar monthly distribution pattern as that of $SO_4^{2-}$. In winter, $NH_4^+$ accounts for about 3% of the $PM_{2.5}$ mass, while in summer for about 12% (Figure 4a, b). The same seasonal pattern for both $SO_4^{2-}$ and $NH_4^+$ has been observed previously in Athens (Mantas et al., 2014; Paraskevopoulou et al., 2015). As in the case of $SO_4^{2-}$, $NH_4^+$ did not present a day to night increase (less than 10% during SP; Table 2).

$NH_4^+$ vs $SO_4^{2-}$ and consequently nss $SO_4^{2-}$, were significantly correlated (r=0.64) for the entire sampling period (December 2013-March 2016), with a slope on a mole basis ($NH_4^+$/nss$SO_4^{2-}$) smaller than unity (0.62), indicating partial neutralisation of nss$SO_4^{2-}$ by $NH_4^+$. This suggests that almost 38% of $SO_4^{2-}$ could potentially be associated with $H^+$ and that a mixture of $NH_4HSO_4$ and $(NH_4)_2SO_4$ is formed in the area. Previous studies in Athens and the Eastern Mediterranean have reached to the same conclusion (Siskos et al., 2001; Bardouki et al., 2003; Koulouri et al., 2008; Theodosi et al., 2011; Paraskevopoulou

et al., 2015). $NH_4^+$ is significantly correlated to $NO_3^-$ only in winter (r=0.73; p<0.001), indicative of $NH_4NO_3$ formation, as previously suggested for the GAA (Karageorgos and Rapsomanikis, 2007; Remoundaki et al., 2013; Paraskevopoulou et al., 2015).

**$NO_3^-$** levels present higher concentrations in the winter (Figures 2b, 5g). This pattern is related to the formation of $NH_4NO_3$ stabilized under the low temperatures prevailing during winter (Park et al., 2005; Mariani and de Mello, 2007). $NO_3^-$ could

originate from local pollution sources, such as vehicular traffic and combustion for heating purposes. $NO_3^-$ levels considerably reduce in summer (Figure 5h), due to the thermal instability and volatilization of the $NH_4NO_3$ (Harrison and Pio, 1983, Querol et al., 2004). Similar seasonal pattern for $NO_3^-$ has been reported previously in Athens (Sillanpää et al., 2006; Paraskevopoulou et al., 2015). During winter months, $NO_3^-$ levels were found to be significantly higher by 53% (53% also if SP is considered; p<0.001 in both cases) during night-time compared to day, indicating important contribution from

heating (Figure 7d, Table 2). A significant correlation of $NO_3^-$ with OC, EC and $PM_{2.5}$ was also observed during winter (r=0.58, 0.60 and 0.56 respectively; p<0.001) further supporting their common origin. In summer no statistically significant correlation between these compounds was found.

For the rest of the ions analyzed, their seasonal distribution depends on their main sources which can be classified into marine, mineral or mixed. $Mg^{2+}$ and $K^+$ are associated with both mineral dust and sea salt aerosol. Commonly, $Cl^-$ has a

major marine influence, as does $Na^+$. nss$Ca^{2+}$ is considered as an effective tracer of crustal sources in the area (Sciare et al., 2005), whereas, nss$K^+$ and $C_2O_4^{2-}$ are considered as tracers of biomass burning (Cachier et al., 1991; Duan et al., 2004; Kawamura et al., 1996; Kawamura and Ikushima, 1993).

**$Cl^-$, $Na^+$ and $Mg^{2+}$** controlled by sea spray emissions are expected to have the same seasonal variability, which is related to the prevailing wind speed and direction. However, the temporal variation of $Mg^{2+}$ concentrations in GAA revealed higher



levels during the warm season (spring-summer), most probably from local dust resuspension and/or regional dust transport, while Cl⁻ and Na⁺ present high levels during winter, most probably due to higher wind speed encountered during this period. Cl⁻ and $Mg^{2+}$ presented no increase (even a slight decrease) during night compared to day, indicating that there might not be a significant influence from wood burning. On the other hand, Na⁺ increased by about 21% (Table 2) indicating a small

contribution from heating and especially biomass burning, as previously reported by Fourtziou et al. (2017).

**nssCa⁺²** is distinctly higher in the warm season due to dust transport from Sahara and/or regional dust resuspension, the latter due to the absence of precipitation. Regarding, **nssK⁺** a bimodal distribution is observed with peaks in spring and winter. The first peak is associated with Saharan dust outbreaks and the second as a result of biomass burning emissions. The latter corroborates well with previous reports from on line fine mode K⁺ measured at the same site (Fourtziou et al., 2017). In

order to discriminate the influence on nssK⁺ levels from Sahara dust vs biomass burning, we have used $Ca^{2+}$ as a tracer of crustal origin. During the period March to October, with no influence from local biomass burning sources, nssK⁺ and $Ca^{2+}$ present significant correlation (r=0.83), confirming their crustal origin. Thus, by using the nssK⁺/$Ca^{2+}$ slope from their linear regression (y=0.82x+0.08), the nssK⁺ of crustal origin ($K^+_{dust}$) can be identified, allowing us to further estimate nssK⁺ of biomass origin ($K^+_{bb}$) from the following equation:

$K^+_{bb} = nssK^+ - K^+_{dust}$ ,

$K^+_{bb}$ levels during the winter period account for 70% of the total nssK⁺ levels and present a well distinguished diurnal cycle with $K^+_{bb}$ concentrations, ranging from 0.5 µg m⁻³ in the night to 0.3 µg m⁻³ in the day, i.e. increase by about 57% (Figure 7e; Table 2; p<0.001), highlighting the role of nssK⁺ as a tracer of wood burning in agreement with Fourtziou et al. (2017). During all three winter campaigns (n>400), the estimated $K^+_{bb}$ correlates significantly with OC (r=0.54), EC (r=0.52) and

NO₃⁻ (r=0.42), especially during night-time (r=0.58, 0.57 and 0.46, respectively) compared to day-time (r=0.15 to 0.23). This result further supports our previous conclusions on the importance of biomass burning on the aforementioned species levels.

$C_2O_4^{2-}$ exhibit peaks during winter and summer, linked to biomass burning emissions and enhanced photochemistry, along with increased biogenic organic compounds' (VOCs) emissions, respectively (Theodosi et al., 2011). $C_2O_4^{2-}$ presents strong correlations with OC and EC during summer (r=0.42-0.63) due to common emission processes such as photochemical and/or

heterogeneous reactions (Myriokefalitakis et al., 2011) $SO_4^{2-}$ presents a significant correlation with $C_2O_4^{2-}$ independent of season (r>0.54; p<0.001). Such correlation has generally been observed in many different sampling locations around the world (Pakkanen et al., 2001; Yao et al., 2003), and can be attributed to heterogeneous reactions during both seasons as proposed by Myriokefalitakis et al. (2011). During winter months, from the compounds impacted by heating sources and examined so far, $C_2O_4^{2-}$ correlates significantly only with NO₃⁻ (r=0.41). In addition, higher concentrations during night

(about 30%) compared to day (Table 2) have been also observed, indicative of the local biomass-burning emissions as important contributors of $C_2O_4^{2-}$. However, the significant correlations with both $SO_4^{2-}$ and tracers of biomass burning clearly indicate that $C_2O_4^{2-}$ have mixed sources of both local (e.g wood burning) and regional origin (e.g. photochemistry) and significant precaution is required when $C_2O_4^{2-}$ is used as exclusive tracer of biomass burning.



### 3.4.4 Trace metals

Trace metals according to their main affinities are classified into crustal-related elements and elements of anthropogenic origin.

### 3.4.4a Crustal – related elements

Al is typically associated with soil dust resuspension as it is linked to natural sources. It presents higher concentrations and larger variations during the transitional (spring and autumn) periods when the air mass trajectories originate predominantly from North Africa and are often associated with intense sporadic peaks of mineral dust. On the other hand, maxima also observed during winter related to road re-suspended dust and lower boundary layer height (Figure 2c). Mn and Fe, which are affected by the mixing of diverse natural and anthropogenic constituents, present the same seasonal variation (Figures 2c, f)
leading to the conclusion that in the GAA they are mainly associated with dust transport. Indeed, Mn reveals a statistically significant correlation with the purely crustal element Al (r=0.59). However, the moderate correlation of Al with Fe (r=0.44), suggests the existence of other sources most probably anthropogenic including resuspension of road dust.

Regarding the diurnal course of Al, it is higher by about 30% during day-time compared to night (Table 2), most probably due to traffic related dust resuspension. Different behavior was observed for the other two "crustal" elements. Mn presented
no difference between day and night-time (Table 2), whereas Fe presented slightly higher levels during night compared to day by about 10% (Table 2). The above described trends, corroborates with our hypothesis for mixed sources, natural and anthropogenic, most probably from heavy oil combustion.

### 3.4.4b Elements of Anthropogenic origin

The analyzed trace metals originating from human activities such as V, Cr, Cd, Ni, Cu, Cd and Pb, relate to a variety of
different sources. In particular, Cr, Cu, Fe, Zn, Cd and Pb are associated with motor vehicle emissions, tire/brake wear debris and industrial origin emissions, while V and Ni with fuel oil combustion. Cr, Mn and Ni, even though they might have a mineral association, they related to industrial and energy-production emissions. Similarly, Cu can be emitted by different processes and despite its industrial origin it is often used as a tracer of non-exhaust vehicle emissions, as in urban areas it originates from traffic related emissions (Weckwerth, 2001; Harrison et al., 2003; Schauer et al., 2006; Amato et al., 2009).
Specifically, in the greater area of Athens it has been identified as an effective tracer for brake wear emissions (Manalis et al., 2005; Grivas et al., 2018). Zn, apart from brake and tire wear, it is included in tailpipe emission due to its presence in lubricating additives (Lough et al., 2005) and Pb is volatilized during braking and re-condensed in the fine fraction (Harrison et al., 2003).

As, Pb and Cd have been proposed as tracers of high temperature processes such as coal combustion amongst other (Pacyna,
1986; Nriagu and Pacyna, 1988; Maenhaut et al., 2016), whilst Cr is usually considered as a tracer for metallurgical activities (Querol et al., 2007), which are actually restricted in the region studied. In fact, As has been associated with wood

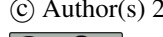



combustion due to the copper chrome arsenate (CCA) treated timber used for residential heating purposes (Fine et al., 2002; Khalil and Rasmussen, 2003; Alastuey et al., 2016).

As presented in Figures 2d-f, the elements of anthropogenic origin exhibit well defined seasonal trends with peak values during winter, as a result of additional sources, especially heating but also meteorology. When their diurnal distribution was examined during winter-time only Cd and Pb presented an evident increase (at 95% confidence level for Cd and Pb) during night-time compared to day, in the range of 11 to 16% (up to about 40% when SP is considered; 99.9% confidence level). The above tendency indicates emission from heating and especially wood burning, in agreement with Maenhaut et al. (2016). The other elements either present an insignificant increase in their levels during night (case of V, Ni) or even a decrease compared to day (case of Cu related to traffic).

By investigating the trace metal inter-correlations, significant correlations of As with Pb and Cd (r=0.39 and 0.66, respectively) were observed. On a seasonal basis, when considering the three intensive winter campaigns higher correlations were obtained between As and Cd (r=0.74) and moderately for Pb (r=0.38), suggesting that heating using coal could be a source of the aforementioned heavy metals (Nava et al., 2015). Strong correlations of Pb with $PM_{2.5}$, OC, EC, $nssK^+$, $NO_3^-$ during winter further reinforce the link with wood combustion sources in our site. Indeed Pb presents significant correlation with $PM_{2.5}$ mass during night-time (r=0.84), as compared to day-time (r=0.14). Interestingly Pb presents the highest correlation with Cd during the summer period (r=0.68), thus indicating regional sources of both elements during that period. Finally, As, Pb and Cd present no pronounced relation to traffic related elements such as Cu and Zn, neither on an annual nor on a seasonally basis.

During summer relatively high correlation coefficients were calculated between the typical heavy oil combustion tracers V and Ni, $SO_4^{2-}$ (r=0.69 and 0.60). This indicates common emission patterns and source types such as shipping. The V/Ni ratio during that period was equal to 1.2, slightly lower than the range of 2 to 4 reported by Viana et al. (2008) to identify shipping emissions, pointing to additional fuel combustion sources. During winter the poorer correlation of V with Ni (r=0.16) can be explained by a decrease in shipping activities and thus Ni could be related to petrochemical and metallurgical activities (see below).

### 3.5 Winter-time $PM_{2.5}$ source apportionment

### 3.5.1 PMF modeling

The model identified six unique factors, characterized as biomass burning, vehicular emissions, regional secondary, heavy oil combustion, dust particles and sea salt. Factors contributions to modelled concentrations of species and source profiles are presented in Figure 8. Contributions of factors to $PM_{2.5}$ concentrations, separately for day-time and night-time sampling periods are shown in Table 3. $PM_{2.5}$ concentrations were adequately reproduced by the model with high correlation coefficient and slope in modeled vs. observed values regression ($r$ = 0.92; slope=0.94; intercept: *n.s.*). Results of the BS-DISP error estimation process indicating the stability of the solution, with over 95% accurate bootstrap mapping, small





change of the Q value and minimal factor swaps. Details are provided in Table S1 according to recommendations of Brown et al. (2015).

### 3.5.2 Source profiles and contributions

**- Biomass Burning (BB)**

The factor is identified by the strong presence of $K^+$ and elevated OC/EC ratios (3.6 on average), suggestive of non-fossil fuel primary emissions. Small amounts of $Cl^-$, $C_2O_4^{2-}$, Fe and Pb were also included (Maenhaut et al., 2016), once again indicating wood-burning associations which have been suggested in section 3.4.4b (Table 2). Ratios of $SO_4^{2-}$ to $K^+$ below unity likely indicate relatively fresh biomass-burning emissions (Viana et al., 2013). Source contributions correlated highly

to the aethalometer-determined $BC_{wb}$ ($r$=0.88). Strong correlations ($r$=0.92) were also observed with the mass fragment of $m/z$ 60, quantified by an aerosol chemical speciation monitor (ACSM) concurrently operating at the same site (Bougiatioti et al., 2014). M/z 60 is considered as a good tracer for biomass burning emissions (Alfarra et al., 2007) and its levels are closely associated with levoglucosan concentrations (Fourtziou et al., 2017). Approximately 35% or EC concentrations are attributed to biomass burning, close to the average winter-time $BC_{wb}$ fraction equal to 32% for the winter period 2015-2016

(Figure 6) and 44% during 2013-2014 (Fourtziou et al. 2017). Similarly, for OC, the PMF estimated BB contribution of 2.7 $\mu g\ m^{-3}$ to fine OC is comparable to 2.3 $\mu g\ m^{-3}$ of biomass-burning organic aerosol estimated to be present in submicron non-refractory OM in Thissio, during the winter of 2013 (Florou et al., 2017).

$NO_3^-$ was predominantly classified along the biomass burning factor. $NO_3^-$ in central Athens is formed as a product of fast chemical processes involving fresh $NO_X$ emissions at a local level (Theodosi et al., 2011). In cold-weather conditions, nitrate

condensation of semi-volatile ammonium nitrate in the particle phase is enhanced. Especially during the night when temperatures drop significantly and fresh wood-burning $NO_X$ are abundant, nitrate concentrations rise significantly (53%; section 3.4.3), establishing a pattern of temporal covariance with biomass burning indicators (Xie et al., 2008; Amato et al., 2016). Moreover, associations between biomass burning aerosols and nitrate that are observed in Southern Greece (Bougiatioti et al., 2014), have been attributed to reduced acidity, which facilitates $NO_3^-$ partitioning in the particle phase

(Guo et al., 2016). Higher night-time pH values are anticipated both due to the ionic content of wood-burning emissions and because of increased water content during the nocturnal hours (Bougiatioti et al., 2016), especially in humid conditions favorable for the occurrence of winter-time smog events.

The average contribution of the factor in $PM_{2.5}$ during night-time is estimated at 3.4 times the daytime value, double the respective increase (1.7 times) in fine aerosol mass (Table 2). Overall, biomass burning was found to be the source with the

largest input to winter-time $PM_{2.5}$ concentrations in the urban-background setting of central Athens. Biomass burning has been recognized as an important contributor to fine particle levels in post economic recession Athens, with mean annual contributions varying between 7-10% (Paraskevopoulou et al., 2015; Amato et al., 2016). Such large wintertime contributions are increasingly being reported for urban background locations in Southern Europe (Nava et al., 2016;



Squizzato et al., 2016; Cesari et al., 2018). Florou et al (2017) have attributed 25% of wintertime (2013) non-refractory submicron aerosol to biomass burning organics at the same site, a result compatible with the presently estimated biomass burning contribution of 32% to $PM_{2.5}$.

**- Fossil fuel sources (VEH and OIL)**

The vehicular emissions factor (VEH) is characterized by an abundance in EC, OC, Cu and to a lesser extent Zn and Pb. Factor contributions correlate well with the fossil fuel fraction of BC ($BC_{ff}$, $r$=0.86). The factor also correlates much better than the biomass burning factor to $NO_X$ ($r$=0.93) and CO ($r$=0.90) concentrations measured at the nearby (0.9 km to the NE) roadside traffic site and considered as indicators of local traffic emissions (respective correlations with BB factor: $r$-

$NO_X$=0.64, $r$-CO=0.63). The EC/OC ratio in the factor equals 0.65, suggestive of vehicular exhaust emissions (Pio et al., 2011) and implying an important input from diesel-powered light- and heavy- duty vehicles in the central area of Athens, in spite of their relatively small presence in the Greek vehicular fleet. The overall contribution of 19% to $PM_{2.5}$ is reasonable for urban background locations in Europe (Belis et al., 2013) and comparable to previous results in Athens (Paraskevopoulou et al., 2015). Night-time contributions of the factors are slightly higher (by a factor of 1.5), however the difference was not

statistically significant at the 0.05 level.

The oil combustion factor (OIL) is dominated by the presence of V and Ni, at ratios indicative of residual oil combustion (V/Ni: 1.8). The observed V/Ni ratio appears to fall short from the typical values reported for shipping emissions (Pandolfi et al., 2011). In Athens, a major part of shipping emissions that affect the inner parts of the basin derive from passenger and cruise ship activity, which during the winter months diminishes. Karageorgos and Rapsomanikis (2010) have reported

wintertime V/Ni ratios of 1.5-1.9 for sites in central Athens for fine particles deriving from mixed harbor and industrial emissions in the S-SW part of Athens. As it can be seen in Figures S1a, b, moderately high contributions are associated with western flows from the industrialized Thriassion plain, while exceedances of the 75[th] percentile are more probable with winds from the harbor zone to the south. No pronounced day-night contrasts were observed, indicating that oil combustion for residential heating should not influence the source profile. The average contribution in the factor is within the range

reported for Mediterranean areas affected by harbor emissions (Perez et al., 2016).

**- Secondary sources (SEC)**

The factor is characterized mainly by sulfate particles. The observed $SO_4^{2-}/NH_4^+$ ratio in the source profile (2.2) is close to the stoichiometric, modified by the presence of $NH_4NO_3$. $C_2O_4^{2-}$ is predominantly associated with this factor, its close

correlation to $SO_4^{2-}$ having been attributed to common in-cloud processing mechanisms (Yu et al, 2005), in agreement with the results in section 3.4.3. The OC/EC ratio exceeds two (2.3). In comparison to year-round observations, reduced OC/EC ratios (Grivas et al., 2012) and regional contributions (Paraskevopoulou et al., 2015) have been documented in the area during winter months, due to limited formation of secondary organics from photo-oxidation processes. Higher contributions were observed during day-time, most probably due to increased photochemical activity. Grivas et al. (2018), estimated 5.5



μg m$^{-3}$ of secondary regional contribution for the cold period of 2011-2012 (extending from mid-October to mid-April), at an urban background location in Central Athens, consistent with values presently reported.

**- Natural Sources (DUST and SS)**

The factor identified as dust is characterized by the presence of Al, Fe, Ca$^{+2}$ and Mn. The observed ratio of Fe/Al (2.1) is higher than the values reported for local top-soil (Argyraki and Kelepertzis, 2014) or for Saharan dust (Formenti et al., 2003). It appears that road dust -rich in trace elements deriving from mechanical wear or vehicles- is incorporated in the dust factor as indicated above in section 3.4.4a. A further indication of the participation of road dust is the abundance of Cr in the factor, which is largely enriched with respect to upper crust composition (EF>100). Contributions do not present significant
day-night variability and overall they account for 8% of PM$_{2.5}$ concentrations, in line with contributions to PM$_{2.5}$ reported for the urban background of Athens (Grivas et al., 2018).

High contributions to Cl$^-$ and Na$^+$ are characteristic for the marine aerosol factor, which records higher-than-median contributions (Figure S1c) mainly during moderate flows from the sea, five km to the S of the site. Ca$^{2+}$ ions participate at a fraction of Cl$^-$ representative of the composition of seawater. Cl$^-$ depletion is limited during the winter months, allowing for a
more realistic quantification of the input of marine aerosols to the fine particle fraction. A relatively higher contribution of the factor to particle mass (9%), in comparison to past studies in Athens is noted (Paraskevopoulou et al., 2015), probably related to the closer vicinity of the site to the sea. An absence of diurnal variability for the source, usually observed in Mediterranean areas and related to mesoscale circulations (Dall'Osto et al., 2012), is reasonable due to the limited evolution of sea-breeze flows in the cool conditions of the measurement period.

**4 Conclusions**

Emissions and emission inventories are core to the effort of reducing the impacts of air pollution episodes. Determining the chemical content of PM$_{2.5}$ atmospheric particles in Athens, in a long term and robust perspective, has been the objective of this study. Source apportionment analysis has also been performed by PMF in order to identify and apportion fine aerosols to their sources. The emphasis on PM$_{2.5}$ is crucial as it represents an essential fraction of particulate matter, closely related to
anthropogenic aerosol and relevant health impacts. This study reports detailed chemical mass closure measurements at Thissio, Athens, covering a 2-year period, from December 2013 to March 2016, including 3 intensive winter campaigns. Approximately 850 daily PM$_{2.5}$ samples were collected and analysed for the main ions, trace metals, OC and EC, identifying a range of useful tracers for monitoring the contribution of the different sources to aerosol load in Athens.

The aerosol chemical mass closure calculations indicated that the relative contributions of the IM, POM, dust, EC and SS are
31%, 34%, 18%, 8% and 3%, respectively. POM and SO$_4^{2-}$ are by far the most abundant species with annual mean concentrations of 6.5±3.2 and 3.0±0.8 μg m$^{-3}$, respectively. Examining their contribution on a seasonal basis, during winter



POM is the dominant component (38%), followed by IM (23%), dust (18%), EC (10%) and SS (4%). In summer the ionic mass is estimated to contribute up to 42%, POM (32%), dust (24%), EC (5%) and SS (4%).

Levels of both POM and EC considerably increased during winter (POM 10.3 µg m$^{-3}$; EC, 2.8 µg m$^{-3}$) compared to summer (POM 4.7 µg m$^{-3}$; EC 0.7 µg m$^{-3}$), underlining the major role of heating-related emissions in Athens during winter. Ionic concentration exhibits a summer maximum, with $SO_4^{2-}$ and $NH_4^+$ concentrations up to 3.8 and 1.7 µg m$^{-3}$, respectively. This is related to the significant contribution from photochemistry during that period, combined with less precipitation and higher regional transport as both compounds are related to regional sources rather than local ones. Given the role of transported mineral dust in the extended region, significant concentration of dust in the PM$_{2.5}$ fraction is observed throughout the year, accounting for 3.6 µg m$^{-3}$ (24%) during summer and 4.9 µg m$^{-3}$ (18%) during winter.

The importance of residential heating was highlighted by examining the diurnal variation of all measured species during winter-time. During the heating period, from November to February, PM$_{2.5}$, POM, EC, $NO_3^-$, nss$K^+$ and $C_2O_4^{2-}$ significantly increased during night compared to day-time, due to intensive combustion of fuel and wood for heating purposes. Heavy metals such as Cd and Pb were also found to be associated to heating activities in winter although in a lesser extend compared to ions and carbonaceous material. In order to further verify the importance of residential heating during winter in the city centre of Athens, PMF was performed with specific emphasis on their day-night contrasts. PM$_{2.5}$ concentrations were adequately reproduced by comparing the modeled versus the observed values ($r$=0.92; slope=0.94;). In total six unique factors were identified during winter characterized as biomass burning, vehicular emissions, heavy oil combustion, regional secondary, marine aerosol and dust particles. Biomass burning was found to be the source with the largest input to winter-time PM$_{2.5}$ concentrations (32%) in the urban-background site of central Athens, with higher night-time contribution (39%) to the PM$_{2.5}$ compared to day-time (19%). The vehicular emissions and oil combustion factors contributed almost equally between night and day, equal to 19% and approximately 7%, respectively for the two factors. The last two factors representing natural emissions (crustal and marine origin) were distinguished with slightly higher contributions during day-time (9% dust and 11% SS) compared to night-time (6% dust and 7% SS). Regional secondary sources were found to be the source with the largest input to winter day-time PM$_{2.5}$ concentrations equal to 30%, higher by a factor of almost 2 compared to night-time contributions (difference statistically significant at the 0.05 level) most probably due to increased photochemical activity.

Consequently, based on these source apportionment results, one can infer that the elevated emissions due to intense anthropogenic activities in the GAA, such as biomass burning, can account for the observed increase in PM$_{2.5}$ levels in winter night-time.

## Acknowledgments

CT and NM acknowledges support by the State Scholarship Foundation ("IKY Fellowships of Excellence for Postgraduate Studies in Greece – Siemens Programme") in the framework of the Hellenic Republic–Siemens Settlement Agreement. EL



and EG acknowledges support from Niarchos Foundation. This study also received financial support from the European Community through the Aerosols, Clouds, and Trace gases Research InfraStructure Network (ACTRIS) Research Infrastructure Action under the 7th Framework Programme (Grant Agreement no 262254).

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





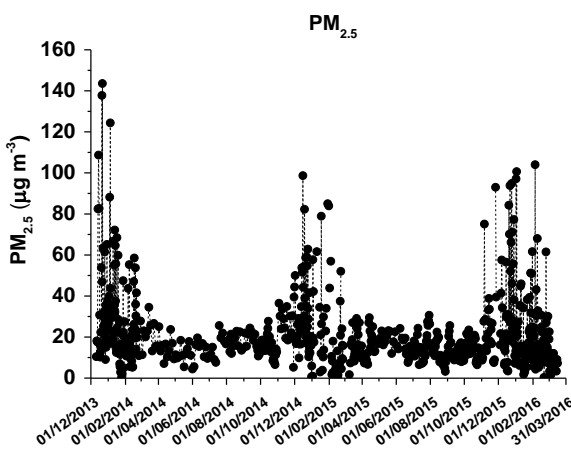

**Figure 1: Daily PM₂.₅ mass concentrations (µg m⁻³) at Thissio station for the studied period December 2013 to March 2016.**

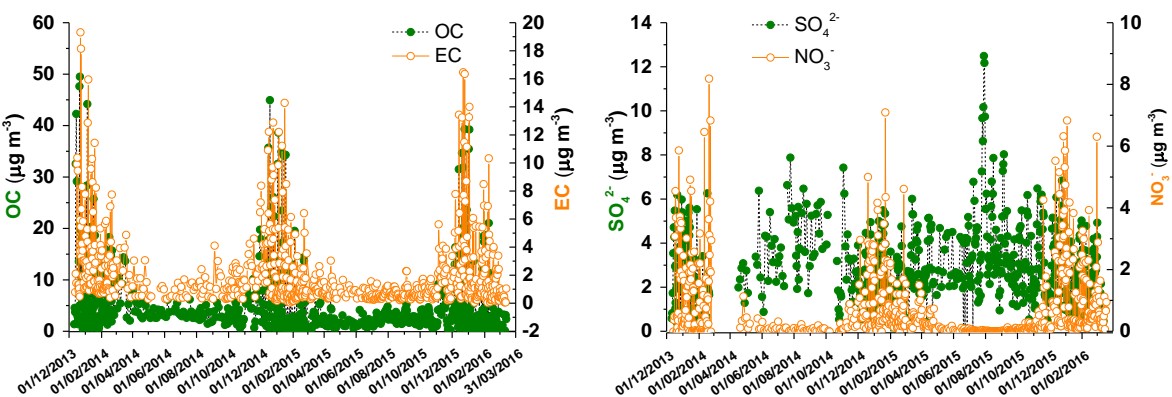



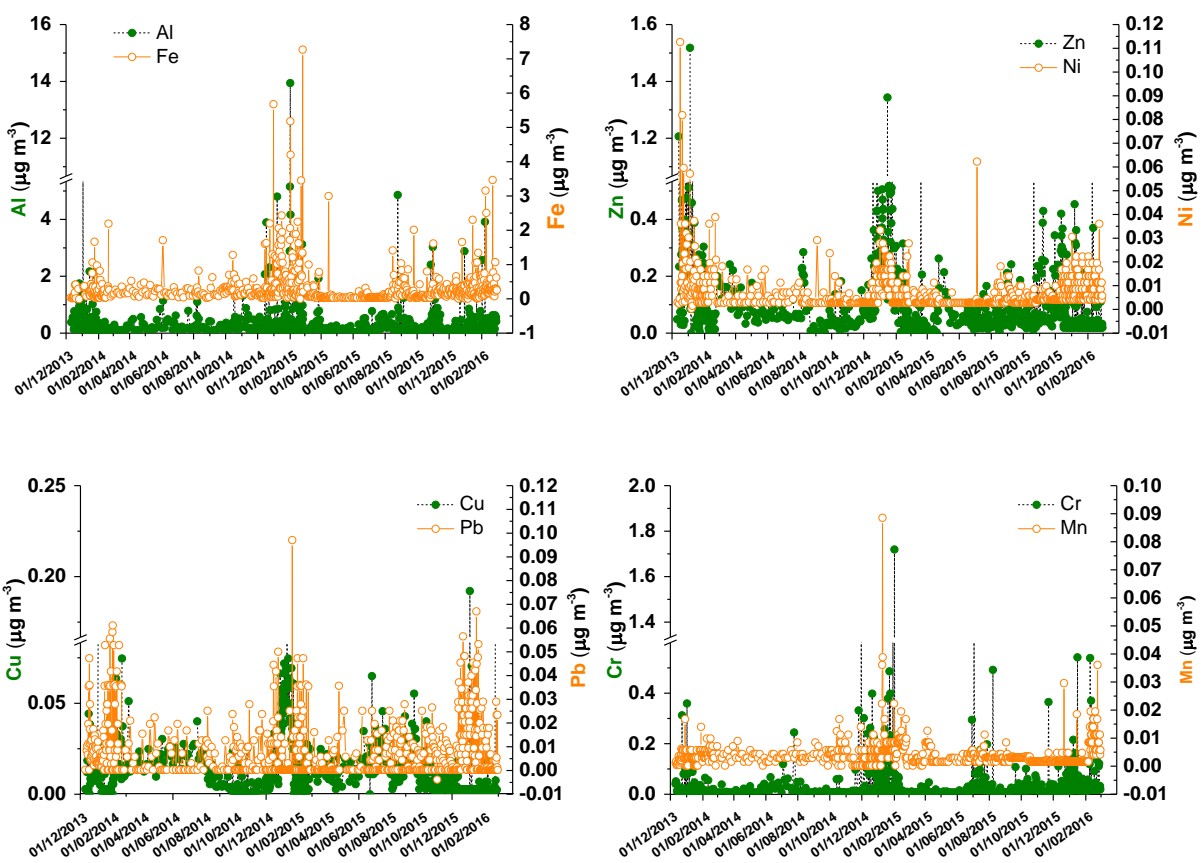

**Figure 2: Daily variation of (a) OC, EC, (b) SO$_4^{2-}$, NO$_3^-$, (c) Al, Fe, (d) Zn, Ni, (e) Cu, Pb and (f) Cr, Mn for PM$_{2.5}$ samples collected at Thissio for the sampling period December 2013–March 2016.**

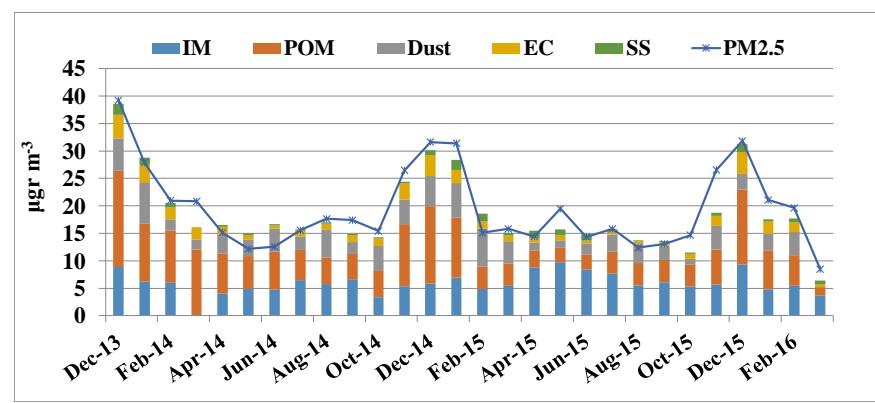

**Figure 3: Annual seasonal chemical mass closure of each aerosol species for PM$_{2.5}$ samples collected at Thissio for the studied period.**



(a)

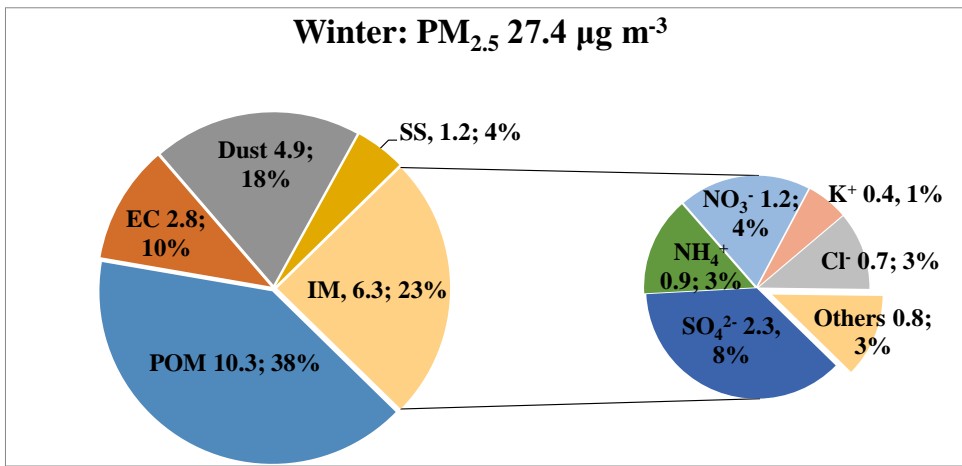

(b)

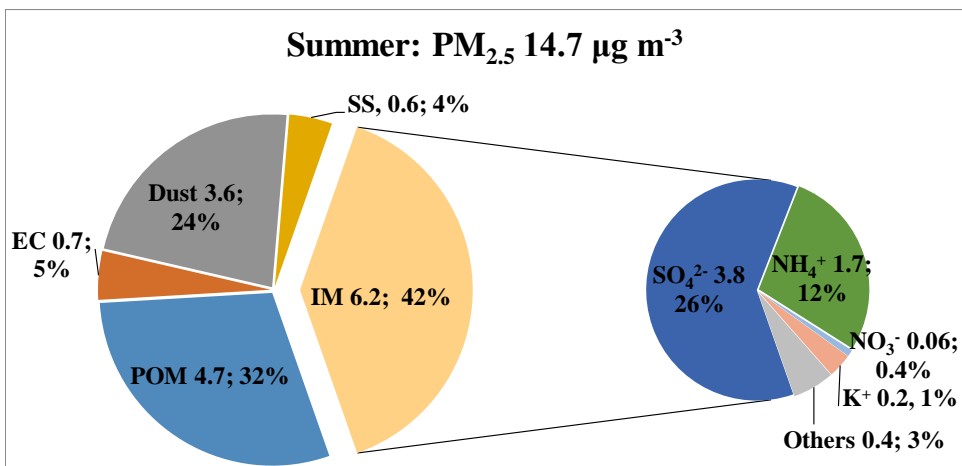

5    (c)

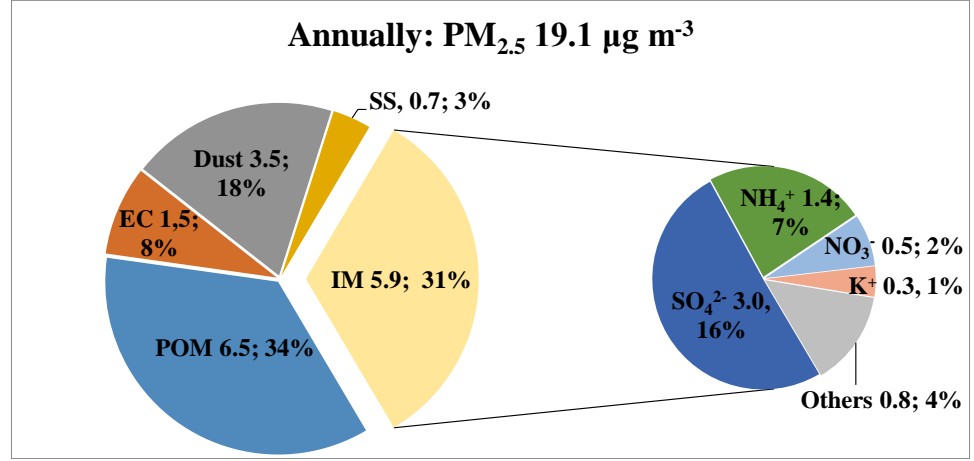




**Figure 4: PM₂.₅ chemical composition and relative contribution of each aerosol species for the studied period (2013–2016) (a) in winter, (b) in summer and (c) annually.**

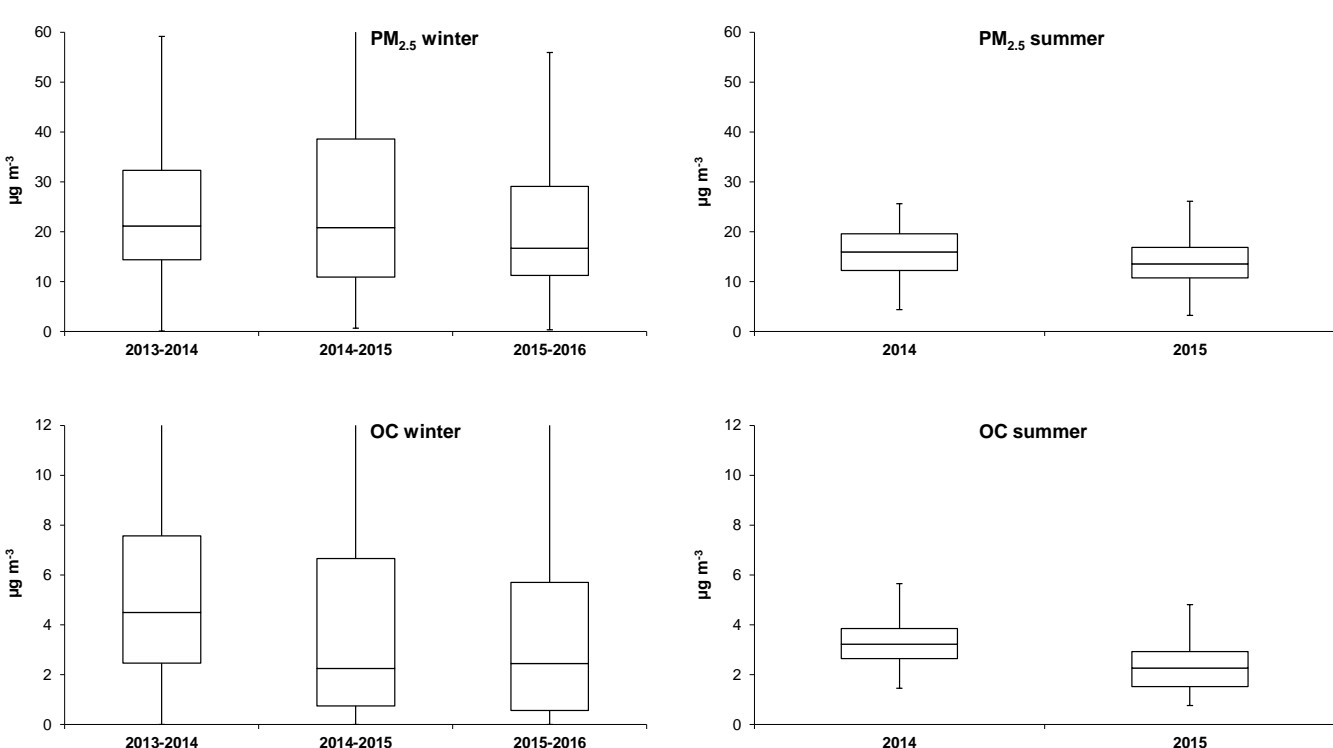




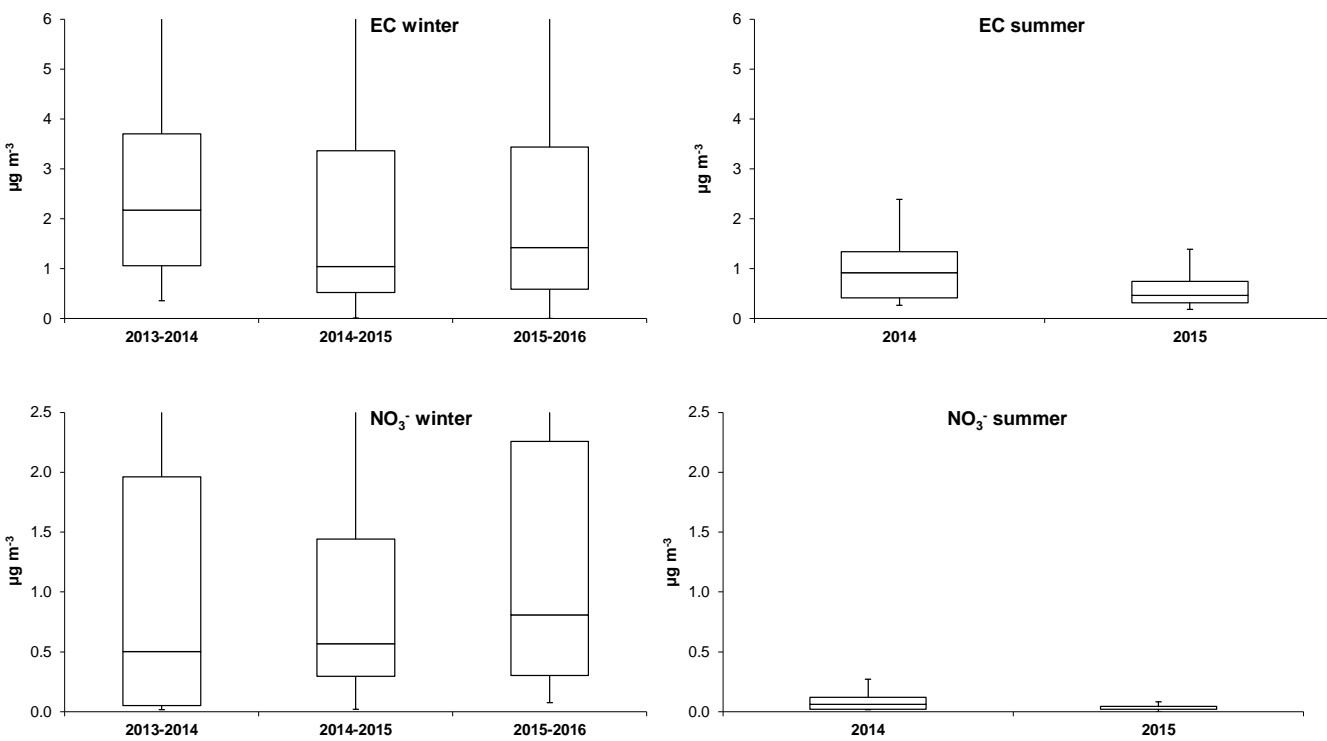

**Figure 5: Winter and summer interquartile range (µg m⁻³) for PM$_{2.5}$ mass, OC, EC and NO$_3^-$ concentrations in the urban site of Thissio for the studied period December 2013 to March 2016.**

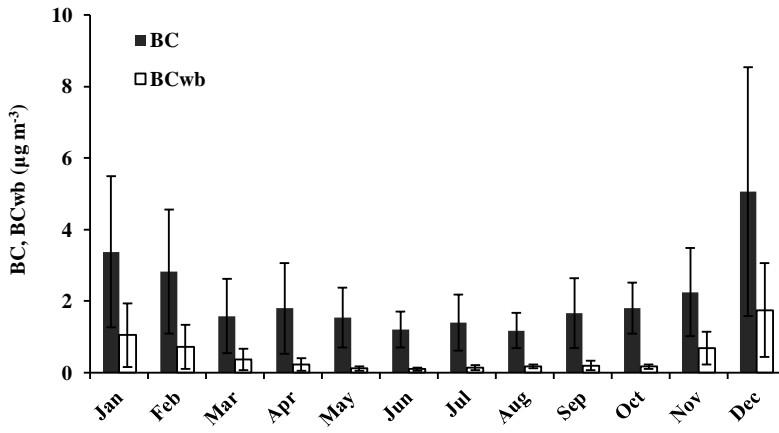

**Figure 6: Seasonal variation of BC and BC$_{wb}$ at Thissio during 2015-2016.**













**Figure 7: Winter PM$_{2.5}$ mass, OC, EC, NO$_3^-$, nssK$^+_{bb}$, As and Pb values (µg m$^{-3}$) divided into day-time and night-time samples. SP refers to concentrations when only smog events occurred.**





**Figure 8: Average contributions to component mass (%, red dots) and source profiles (μg mg⁻¹, colored bars) of PMF-resolved sources. Error bars providing the interquartile range from bootstrap resamples.**




| ng m$^{-3}$ | PM$_{2.5}$ | | PM$_{2.5}$ | | PM$_{2.5}$ | PM$_{2.5}$ | PM$_{2.5}$ |
| --- | --- | --- | --- | --- | --- | --- | --- |
| | Mean | Median | Mean | range | Summer/Winter Mean | Mean±stdev | Mean±stdev |
| PM$_{2.5}$* | 19.1±7.1 | 15.5 | 80.7 | 27-127 | | 29.4±10.3 | 33±12 |
| OC* | 4.0±2.0 | 3.12 | 16.9 | 5.7-43.4 | 25.7/16.1 | - | 5.55±2.01 |
| EC* | 1.5±1.0 | 1.12 | 4.2 | 1.2-18.6 | 8.20/11.0 | - | 2.06±0.78 |
| Cl$^-$ | 272±197 | 155 | 350 | 0-1300 | 75/440** | 490 | 182±80.2 |
| Br$^-$ | 19.4±10.7 | 14.2 | 130 | 0-589 | 140/380** | - | - |
| NO$_3^-$ | 476±377 | 278 | 2000 | 500-5100 | - | 1090 | 896±634 |
| HPO$_4^{2-}$ | 54.3±37.4 | 46.4 | - | - | - | - | 72.9±73.6 |
| SO$_4^{2-}$ | 3033±835 | 2933 | 10400 | 4100-23700 | - | 5790 | 3293±1473 |
| C$_2$O$_4^{2-}$ | 143±56 | 123 | - | - | - | 320 | 203±50.4 |
| Na$^+$ | 307±196 | 369 | 369 | 0-1890 | - | 830 | 255±78.7 |
| NH$_4^+$ | 1422±685 | 1112 | - | - | - | 920 | 843±433 |
| K$^+$ | 273±109 | 268 | 428 | 24.2-833 | 160/380** | 460 | 150±76.7 |
| Mg$^{2+}$ | 64.0±31.6 | 60.7 | 132 | 0-653 | - | 50 | 8.84±10.9 |
| Ca$^{2+}$ | 160±127 | 113 | - | - | - | 320 | 219±246 |
| Al | 293±106 | 263 | 408 | - | 130/850 | - | 494±95.3 |
| As | 1.9±0.2 | 1.83 | - | - | 34/13 | - | 4.35±5.56 |
| Ca | 745±522 | 501 | 2610 | 776-4970 | - | - | 898±254 |
| Cd | 1.7±0.7 | 1.75 | 220 | 0-1141 | - | - | 1.16±0.60 |
| Cr | 27.1±20.4 | 19.6 | - | - | - | - | 18.8±6.29 |
| Cu | 14.8±5.2 | 16.2 | 50.6 | 0-143 | 35/260 | - | 17.4±5.75 |
| Fe | 264±125 | 234 | 1100 | 293-1990 | 490/300 | - | 304±110 |
| V | 6.9±2.4 | 6.61 | - | - | -/6.60 | - | 7.79±3.86 |
| Zn | 93.0±50.1 | 73.1 | 273 | 46.8-615 | 130/210 | - | 94.9±34.8 |
| Mn | 4.6±1.4 | 5.06 | 21.6 | 0-70.3 | -/14.0 | - | 5.68±1.23 |
| Ni | 6.0±2.6 | 4.86 | 11.6 | 0-37.4 | 21.0/8.60 | - | 5.16±2.41 |
| Pb | 4.1±3.4 | 3.68 | 699 | 162-2273 | 610/1100 | - | 12.5±2.46 |
| Sampling | Dec 2013-March 2016 (n=850) | | Summer 1987 (n=27) | | Summer 1982 /winter 1982-1983 (n=29) | September 2005-August 2006 (n=109) | 4 short- term seasonal campaigns June 2011 through February 2013 (n=211) |
| Location | Athens urban | | Athens urban | | Athens urban | Athens urban | Athens urban |
| Reference | This study | | Scheff and Valiozis, 1990 | | Valaoras et al. (1988) | Theodosi et al. (2011) | Paraskevopoulou et al. (2015) |



* µg m$^{-3}$, ** elemental concentrations

**Table 1: The annual mean, standard deviation, median and range of measured concentrations for PM$_{2.5}$ aerosol samples, collected at Thissio and other urban sites in Athens.**

| % | Full winter period | | | Only SP | | |
|---|---|---|---|---|---|---|
| | mean | stdev | median | mean | stdev | median |
| PM$_{2.5}$ | 80%**** | 120% | 55% | 96%**** | 158% | 80% |
| OC | 254%**** | 278% | 200% | 278%**** | 285% | 232% |
| EC | 134%**** | 189% | 113% | 148%**** | 207% | 115% |
| Cl$^-$ | -6% | 2% | -6% | -1% | 11% | -10% |
| NO$_3^-$ | 53%**** | 32% | 90% | 53%**** | 17% | 111% |
| SO$_4^{2-}$ | 9% | 15% | 3% | 11%**** | 14% | 7% |
| C$_2$O$_4^{2-}$ | 29%*** | 18% | 33% | 28%* | 22% | 35% |
| Na$^+$ | 36% | 219% | 6% | 21% | 46% | 13% |
| NH$_4^+$ | 13%**** | 7% | 13% | 9%**** | 5% | 12% |
| nssK$^+_{bb}$ | 54%**** | 68% | 34% | 57%*** | 53% | 39% |
| Mg$^{2+}$ | -11% | 16% | 0% | 0% | 13% | 0% |
| Ca$^{2+}$ | -57% | 65% | 0% | -70% | 90% | 0% |
| Al | -26% | 53% | -19% | -11% | 40% | -19% |
| As | 11% | 2% | 26% | 22% | 4% | 70% |
| Cd | 16%* | 6% | 0% | 37%**** | 12% | 0% |
| Cr | -2% | 20% | -13% | 21% | 15% | 33% |
| Cu | -24% | 66% | -50% | -23% | 70% | -50% |
| Fe | 9% | 10% | 13% | 31%* | 5% | 29% |
| V | 1% | 3% | 0% | 5% | 4% | 17% |
| Zn | 18% | 34% | 9% | 9% | 4% | 30% |
| Mn | -5% | 27% | 0% | 1% | 14% | 0% |
| Ni | -3% | 17% | 0% | 5% | 7% | 33% |
| Pb | 12%* | 35% | 10% | 20%**** | 34% | 19% |

****$p < 0.001$ (99.9%) *** $p < 0.01$ (99%), ** $p < 0.025$ (97.5%), * $p < 0.05$ (95%)

**Table 2: % increase in the diurnal distribution of all studied elements during night-time compared to day in winter for all winter samples (n=447) and smog pollution events (SP, n=289).**

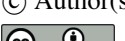



| Source | Day | Night | Significance ($p$) |
|---|---|---|---|
| Biomass Burning | 3.4 (19) | 11.4 (39) | 0.00* |
| Vehicular | 3.4 (19) | 5.4 (19) | 0.11 |
| Secondary | 5.3 (30) | 4.1 (14) | 0.00* |
| Oil Combustion | 1.4 (8) | 1.6 (6) | 0.52 |
| Dust | 1.6 (9) | 1.8 (6) | 0.57 |
| Sea Salt | 2.1 (11) | 2.1 (7) | 0.18 |
| *Unaccounted* | 0.4 (2) | 2.6 (9) | |

*significant at the 0.05 level

**Table 3: Average day/night contributions of identified sources to PM$_{2.5}$ (µg m$^{-3}$). Respective percentages in parentheses. Statistical significance (p) refers to differences for day-night pairwise comparison assessed with Wilcoxon signed-rank non-parametric tests.**

