# Peer review of "Multiyear chemical composition of the fine aerosol fraction in Athens, Greece, with emphasis on the contribution of residential heating in winter-time"

_Atmospheric Chemistry and Physics, 2018_

## Referee Comment (RC1) · Anonymous Referee #2 · 1 May 2018

The manuscript presents and discusses results from a long-term study of the PM2.5 aerosol composition for Athens, Greece. Emphasis is given to the impact of residential heating during winter. The manuscript is definitely of interest to aerosol researchers who are involved in related work. The study would have benefited from measuring the wood-burning marker levoglucosan during the winter campaigns, though. Since no pure wood burning marker was included in the analysis, the assessment by PMF of the contribution from wood burning has likely a substantial associated uncertainty. There is no information given about this uncertainty in the manuscript, but there should be.

Furthermore, as indicated below, the current manuscript is on several occasions un-

[Figure]

clear, there are problems with the references, and the manuscript also suffers from many other (mostly technical) shortcomings. Consequently, substantial revision is needed before it can be published in ACP.

Specific comments:

1. Page 2, line 24: It seems that there is something missing after "between winter 2012".

2. Page 2, line 12: Is there something missing after "the city center and" or should the "and" just be deleted?

3. Page 3, lines 25-29, and further within the text: It seems that during winter separate day and night samples were collected in addition to the regular 24-h samples. However, later in the manuscript (e.g., in Figure 4(a) and Figure 5), it is not clear whether the winter data were derived from the regular 24-h samples or from the day and night samples.

4. Page 4, line 26: Abbreviations and acronyms (here PMF) should be defined (written full-out) when used for the first time within the body of the manuscript.

5. Page 7, line 23: Abbreviations and acronyms (here IM and SS) should be defined (written full-out) when used for the first time within the body of the manuscript.

6. Page 8, lines 1-2: There are no sulphate and ammonium data in Figure 3.

7. Page 8, line 8: It is unclear whether the number of samples (780) includes only 24-h samples or also day and/or night samples.

8. Page 8, line 15: It is unclear whether the winter PM2.5 mass was derived from the 24-h samples or from the day and night samples.

9. Page 8, line 16, and further in the manuscript: Reference is made here to Figures 5a,b and later to subfigures for Figure 7, but the different subfigures of Figures 5 and 7 are not labeled with a, b, c, and so on.

10. Page 8, lines 22-23: There is no info on the aethalometer in Section 2. What size fraction was measured with it? And how was BCwb obtained? Some info and/or literature references on this are needed in Section 2.

11. Page 9, line 16: The use of the word "between" is incorrect here. Could it be replaced by "of"?

12. Page 9, line 17: Why is the word "morning" used here? How were "morning" data obtained? Should it not be "day"?

13. Page 9, line 27: Why is there only one correlation coefficient given for the winter? Or were the same correlations obtained for OC and EC?

14. Page 10, lines 10-13: I do not understand the reasoning. If the molar ratio of NH4/nssSO4 is smaller than unity, would one not expect a mixture of H2SO4 and NH4HSO4?

15. Page 11, line 17: It is unclear what is meant by "increase by about 57%". Increase of what relative to what?

16. Page 13, lines 11-13: I cannot follow the reasoning. Would the correlations between the three elements during winter not suggest that wood burning is an important source, considering that this was also observed in the study of Maenhaut et al. (2016)?

17. Page 16, line 9: Abbreviations and acronyms (here EF) should be defined (written full-out) when used for the first time within the manuscript.

18. Page 17, line 12: It is unclear what is meant by "fuel" here.

19. Pages 18-26, Reference list: There are several problems. For authors with more than 1 initial, there should be a space between the initials; each initial should be followed by a full stop (.); for references with at least 3 authors, there should be ", and" before the last author (note that for references with only 2 authors, there should not be ", and" but " and" instead; the publication year should be at the end of the reference

and not after the authors; "Kawamura and Iskushima, 1993" (on page 21, lines 13-14) should be moved down to before "Kawamura et al., 1996"; "Uria-Tellaexte and Carslaw, 2014" (on page 25, lines 9-10) should be moved down to before "Valaoras et al., 1998".

20. Further problems with the references: The following references are in the text, but not in the Reference list:

- page 14, lines 22 and 32: Amato et al., 2016.

- page 16, line 18: Dall'Osto et al., 2012; there is Dall'Osto et al., 2013 in the Reference list to which not is referred within the text.

21. Page 31, caption of Figure 5: This caption should be extended. What does the line inside each box indicate? And what do the whiskers mean? Why do some whiskers end on a dash and other not?

22. Technical and other (mostly minor) corrections:

- page 2, line 2: replace "interest on" by "interest in".

- page 2, lines 3 and 20; page 8, line 24; page 11, line 32: replace "e.g." by "e.g.,".

- page 2, line 13, and on several occasions further in the manuscript: for references within parentheses there should be a comma before the publication year.

- page 2, line 30; page 4, line 2; page 7, line 3: replace "water soluble" by "water-soluble".

- page 2, line 33: replace "in the" by "in".

- page 3, line 8: replace "Athens," by "Athens".

- page 3, line 16; page 10, line 4: replace "long range" by "long-range".

- page 3, line 17: replace "Northern sector" by "northern sector".

- page 3, line 19: replace "Southern sector" by "southern sector".

- page 3, line 21: replace "Athens" by "Athens'".

- page 3, line 22: replace "are provided" by "is provided'".

- page 3, line 27: replace "to the" by "in the'".

- page 4, line 11: replace "species concentrations" by "species".

- page 4, line 13: replace "in ultrasonic bath and" by "in an ultrasonic bath and the extracts were" and replace "0.45$\mu$m" by "0.45 $\mu$m".

- page 4, line 17: replace "in details" by "in detail".

- page 4, line 20: replace "while" by "while for".

- page 4, line 27: replace "on 12-h" by "on the 12-h".

- page 4, line 31: replace "with analysis" by "with the analysis".

- page 5, line 10: replace "their contributions" by "the day and night contributions".

- page 5, lines 17-18: replace "probability (CBPF) function" by "probability function (CBPF)".

- page 5, line 29: replace "concentration of" by "concentrations of".

- page 6, line 8: replace "represents time" by "represents the time".

- page 6, line 22: replace "concentration of" by "concentrations of".

- page 7, line 14: replace "confined in" by "confined in the" and replace "they do not" by "do not".

- page 7, line 15: replace "exceed few" by "exceed a few".

- page 7, line 17: replace "trace element" by "the trace element".

- page 7, line 23: replace "Mass Closure" by "mass closure".

- page 7, line 27: replace "relative ratio" by "concentration".
- page 8, line 19: replace "low winds" by "low wind speeds".
- page 8, line 24: replace "in GAA" by "in the GAA".
- page 8, line 30: replace "i.e" by "i.e.,".
- page 8, line 32: replace "have been" by "has been".
- page 9, line 8: replace "winter POM" by "winter, POM".
- page 9, line 12: replace "on carbonaceous" by "on the carbonaceous".
- page 9, line 13: replace "Similar" by "A similar".
- page 9, line 19: replace "on fine" by "on the fine".
- page 9, line 26: replace "correlation with" by "correlation with the".
- page 9, line 28: replace "Composition" by "composition".
- page 10, line 1: replace "low winds" by "low wind speeds".
- page 10, line 21: replace "considerably reduce" by "are considerably reduced".
- page 10, line 22: replace "Similar" by "A similar".
- page 10, line 34: replace "concentrations in" by "in".
- page 11, line 6: replace "from Sahara" by "from the Sahara".
- page 11, line 7: replace "Regarding, nssK+" by "Regarding nssK+,".
- page 11, line 9: replace "from on line" by "from online".
- page 11, line 12: replace "present" by "exhibit a".
- page 11, line 17: replace "i.e." by "i.e.,".
- page 11, line 22: replace "exhibit" by "exhibits".

- page 11, line 23: delete ", respectively".

- page 11, line 25: replace "2011)" by "2011).".

- page 11, line 32: replace "e.g wood" by "e.g., wood".

- page 12, line 5: replace "as it is linked" by "and thus mainly linked".

- page 12, line 7: replace "maxima also" by "maxima are also".

- page 12, line 8: replace "road re-suspended" by "resuspended road".

- page 12, line 14: replace "Different" by "A different".

- page 12, line 16: replace "trends, corroborates" by "trends corroborate".

- page 12, line 18: replace "Anthropogenic" by "anthropogenic".

- page 12, line 19: replace "The analyzed" by "The measured".

- page 12, line 22: replace "they related" by "are related".

- page 12, line 26: replace "it is included" by "is included".

- page 12, line 29: replace "amongst other" by "and non-ferrous metals production amongst others".

- page 13, line 14: replace "in our site" by "at our site" and replace "significant" by "a significant".

- page 13, line 18: replace "seasonally basis" by "seasonal basis".

- page 13, line 28: replace "Factors" by "Factor".

- page 13, line 30: replace "PM2.5" by "The PM2.5".

- page 13, line 31: replace "n.s." by "not significantly different from 0".

- page 13, line 32: replace "indicating" by "indicated" and place "indicated the stability

of the solution" at the end of the sentence on page 14, line 1.

- page 14, line 5: replace "Burning" by "burning".

- page 14, line 13: replace "or EC" by "of EC".

- page 14, line 16: replace "to 2.3" by "to the 2.3".

- page 14, line 21: replace "are abundant" by "is abundant".

- page 15, line 14: replace ", however" by "; however,".

- page 16, line 4: replace "Sources" by "sources".

- page 16, line 9: replace "to upper" by "to the upper".

- page 16, line 10: replace "8% of" by "8% of the".

- page 16, line 28: replace "to aerosol" by "to the aerosol".

- page 17, line 2: the parentheses around the percentages should be removed.

- page 17, line 4: replace "Ionic" by "The ionic".

- page 17, line 8: replace "significant" by "a significant".

- page 17, line 13: replace "lesser extend" by "lesser extent".

- page 17, line 19: replace "with higher" by "with a higher".

- page 17, line 21: replace "respectively" by "respectively,".

- page 17, line 31, and page 18, line 1: replace "acknowledges" by "acknowledge".

- page 23, line 7: replace "Nair, and" by "and Nair,".

- page 25, line 27: replace "Waenhaut" by "Maenhaut".

- page 30, line 1: replace "aerosol species" by "aerosol component or species".

- page 35, line 1: replace "to component" by "to the component".

- pages 36, 37 and 38: the Table headings should be above the Tables instead of below.

- page 36: the information on the Sampling, Location and Reference should be at the top of Table 1 instead of at the bottom.

- page 37, line 3: replace "The annual" by "Annual".

- page 37, line 27: replace "elements during" by "elements and species during".

Comments for the Supplement:

Page 2: It is unclear what "CPF probability" indicates. In the Main text CBPF is mentioned.

Page 3, line 1: Replace "wind speed and speed" by "wind direction and speed in m s-1".

Page 4: The Table heading should be above the Table instead of below.

---

## Referee Comment (RC2) · Anonymous Referee #3 · 7 May 2018

A long data series of PM2.5 speciation (more than 800 samples) obtained at an urban background site in Athens during a two-year period is presented. Description is focused on the winter period in order to quantify the impact of heating emissions and to define effective mitigation actions. A Positive Matrix Factorization was applied, identifying six sources for PM. The importance of domestic heating in winter is highlighted, and authors suggested a significant contribution of biomass burning (BB). However, this contribution can be overestimated (see below). Results obtained with an aethalometer are used for the interpretation of the BB contribution. However, these measurements are not properly described. The text is long and redundant. An effort must be made to reduce the length of the text Some minor clarifications are needed before publication

in ACP Page 3, section 2.2. Please, clarify the analysis performed for the 24h and the 12h samples. Were the 24h and 12h filters collected simultaneously during winter? Did you analyze total K, Mg and Na by ICP? Only the soluble fraction of these elements is presented, However, the contribution of the insoluble fraction of these elements can be important during dust events. Section 2.3: did you consider both, 24h and 12h filters for the PMF? Why did you perform the PMF analysis for the winter samples only? A higher number of cases may favor the identification of factors. Section 3.1 and 3.2. There are more recent papers on PM2.5 levels and composition and source apportionment in Europe. See Amato et al., Atmos. Chem. Phys., 16, 3289-3309, 2016 Section 3.2.1. Please, use ranges from <DL (specify for each component) to the maximum value; avoid referring to 0 concentrations. Page 7; line 12. These elements (Mn, Cr, Ni, V, As...) may also be associated to mineral dust. I would say "...of major anthropogenic origin..." Section 3.3; page 7, line 31. How did you estimate the sea salt fraction of these PM components? Please, add references. Page 8, section 3.4.1. The combined effect of meteorology (less developed boundary layer, low wind speed...) and the increase of the domestic emissions result in a poor air quality in winter. Thus, page 8 – line 28-30; the increase of PM2.5 concentrations during the night in winter may be related to the lower development of the boundary layer during the night and not only to the increase of the emissions of domestic heating. More information on the aethalometer measurements and model is needed . The description of the BC measurements should be included in the section 3.4.2 on carbonaceous components. Does the daily evolution of BCwb differ from that of BCff? Page 10. Lines 28-32. The insoluble fraction of Mg, K and Na is usually associated to the dust fraction. Page 11, lines 10 to 15. Mineral K is usually associated to clay minerals more than to carbonates. Could you check the correlation between K+ and Al for dust events and for non-biomass periods? Page 11, line 25. Please, replace ...2011)" by ...2011)." Page 14. Section 3.5.2 Line10. Please, add a figure or a reference for BB vs BCwb correlation. Line 11. Please, specify the period for the ACSM measurements carried out by Bougiatioti et al., 2014. The PMF profile for BB is characterized by a high contribution of nitrates.

This correlation can be due to covariation induced by meteorological factors Authors explained the association of nitrates with the BB factor by high emission of NOx by biomass burning and by the lower acidity. However, as explained later NOx better correlated with the vehicular emission factor. May a fraction of nitrates be derived from fuel oil emissions? Then, the contribution of BB may be overestimated. Page 15. The vehicular source is characterized by a relatively higher contribution during night in the winter period. Can be this diurnal trend related to a significant contribution of fuel oil domestic heating? The vehicular source highly correlated with BCff; does the diurnal trend of BCff reproduce the trend of vehicular traffic? Is there other source of BCff during night in winter? Page 15, Line 20. V/Ni derived from fuel oil combustion sources is usually associated to SO42-; however, SO42- is not significantly associated in this factor

---

## Referee Comment (RC3) · Anonymous Referee #1 · 22 May 2018

The paper by Theodosi et al. reports about the large observational study monitoring particulate matter sources in central Athens emphasizing contribution of biomass burning, especially during winter and in particular during the night. The study presents no new knowledge, nor scientific advancement, although it provides very detail account of sources and is methodologically correct. The paper is often buried in details and often repetitive and better focus on the main idea is necessary. At the moment the paper looks more like a report than scientific paper. Better focus and summary at the end of sections is necessary and perhaps separating results from discussion. The paper ends abruptly with conclusions as an extensive summary. Quite an improvement (including the English) is needed before the paper can be accepted for publication.

[Figure]

Comments:

I guess the authors are overstretching when repeatedly refer to the Greater Athens Area, because a single station in the city centre can hardly described the entire Greater Athens area. Clearly more stations are necessary to monitor greater agglomerate area due to different population/build up density, anthropogenic activities and traffic. Therefore, a better reasoning is required to refer to GAA if possible at all.

Very often the authors go into unnecessary simplification and repetition, e.g. regarding temperature inversions and limited mixing. Every specialist is aware of the phenomenon and mentioning it once in detail is enough.

Minor comments:

Page 2, line 6. through emission reduction measures.

Line 9. . . .toxic and carcinogenic components.

Line 12. Trace metals are also related to chronic and acute...

Line 18. . . .such as traffic and industrial activities.

Line 19. since the winter 2011-2012.

Line 21. the great impact.

Line 25. Linking them to the presence

Line 33. . . .has been undertaken in Southern Europe, offering challenging conditions (what is challenging there by the way?)

Page 3, line 2. the current work was focused on winter period.

Line 4. The aerosol sources during the night are not new. Rephrase to "... highlight the impact of night time PM sources".

Line 10. It is still representative of central Athens, especially during high pollution

events during stagnant meteorological conditions. A single site cannot represent the whole GAA as suburban areas were not monitored. It is methodologically wrong to assume that a single site can serve as a reliable average of the entire GAA.

Line 13. why this word of caution when the next sentence explains everything?

Line 17. On a yearly basis, air masses of Northern origin from central and eastern Europe account for almost two thirds of the time.

Line 20. Ventilation is a poor term. You should be stating about stagnant conditions during which most severe pollution events have been occurred persisted for X % of time.

Line 28. Unclear. Were the 447 samples in addition to 848 samples or a fraction of them?

Line 30. Controlled RH=%?? conditions. Please specify.

Page 4, line 17. ...in detail by Theodosi...

Line 26. Not contrast, but patterns, as no one can know in advance if they will contrast.

Line 27. trivial repetitive sentence.

Page 5, line 9. unclear - combined OR night-day?

Line 27. Was it really lower limit at 0.3ug/m3 which is inconceivably low for urban PM2.5 concentration. Was it realistic and how did it compare to chemical mass?

Page 6, line 8. zero OC or EC cannot be observed anywhere on Earth (even in Antarctica there is observable BC of ~1ng/m3). Refer to below DL of Xug/m3.

Line 10. To be mathematically correct one should only present the arithmetic average if data are normally distributed in which case arithmetic average and median are closely similar number. We don't see that therefore data are lognormally distributed in which case the range and the median should be presented only. It has to be consistently

presented either median alone or ALWAYS with the median, rather then selectively presenting median.

Line 26. Excellent agreement should not be necessarily expected, "results were in close agreement". Close agreement suggests long-range transport as the main source.

Page 7, line 17. remained within the same order of magnitude between studies with only slight differences in between. Chemical mass closure. Given that all major and minor particulate matter species were measured, the chemical mass closure has been attempted. However, given the large uncertainty in OM/OC ratio and no data for the current study, how useful the chemical closure exercise was?

Page 8, lin 19. Could further limit dispersion of pollutants.

Line 31. 289 days with smog conditions ...

Page 9, line 29. No statistically significant seasonality can be seen in SO4 time series considering error bars. Note the lowest concentrations in November-December 2014. Rewrite as the text is not reflecting the Figure.

Page 10, line 4. What is the anthropogenic source of SO4 during summer if long-range transport is blamed? Is it not biogenic (Mediterranean DMS) instead?

Page 11, line 3. ...suggesting insignificant influence from wood burning. Is it impossible that fraction of Na increase is due to higher primary in winter?

Page 12. 3.4.4a Crustal elements

Line 26. Present in tailpipe emissions, not included.

Page 14. Lines 31-34. Reword – confusion with mean annual contributions and "such large wintertime contributions".

Page 15, line 13. What is the reason for heavy duty vehicles in central Athens?

Line 23. westerly advections, not flows.

[Figure]

Line 28. there is no information whether particles were internally or externally mixed. And even if known there is no sulphate particle in existence.

---

## Editor Comment (EC1) · X. Querol (Editor) · 8 Jun 2018

Dear authors,

When replying to the referees' comments, please be also clear in showing what are the differences on the major scientific objectives of this paper and the one being discussed now in https://www.atmos-chem-phys-discuss.net/acp-2018-356/

Kind regards

Xavier

---

## Author Comment (AC1) · 26 Jun 2018

*The manuscript presents and discusses results from a long-term study of the PM2.5 aerosol composition for Athens, Greece. Emphasis is given to the impact of residential heating during winter. The manuscript is definitely of interest to aerosol researchers who are involved in related work. The study would have benefited from measuring the wood-burning marker levoglucosan during the winter campaigns, though. Since no pure wood burning marker was included in the analysis, the assessment by PMF of the contribution from wood burning has likely a substantial associated uncertainty. There is no information given about this uncertainty in the manuscript, but there should be. Furthermore, as indicated below, the current manuscript is on several occasions unclear, there are problems with the references, and the manuscript also suffers from many other (mostly technical) shortcomings. Consequently, substantial revision is needed before it can be published in ACP.*

We would like to thank the reviewer for his/her comments. We agree that a specific biomass burning tracer other than soluble K would have strengthened the characterization of the relevant factor. However, due to the large number of filters in the study it was not possible to perform levoclugosan measurements as well. Given the correlations of the factor with the biomass burning tracers (BCwb and $m/z$ 60), component ratios and day-night variability, we believe that the factor is adequately characterized. However, the reviewer is right to indicate a likely uncertainty and this is now reported in the revised manuscript.

Furthermore, in order to provide an indication of such an uncertainty, we have repeated the PMF analysis including data on the $m/z$ 60 and $m/z$ 73 fragments, from collocated ACSM measurements, for the winter of 2015-2016, when such data were available. From previous work conducted at the present site (Fourtziou et al.,2017) these have been validated as important BB tracers and are considered fingerprint fragments of levoglucosan, displaying strong correlations.

The $m/z$ 60 and 73 values in $\mu g\ m^{-3}$ were averaged for the respective filter sampling intervals. The number of available cases was 109, still adequate to perform the analysis. We note that combining data from chemical analysis and aerosol mass spectrometry in the same receptor model, while not frequent, has been attempted in past studies (Li et al., 2004; Dall' Osto et al., 2014).

A similar 6-factor solution was again obtained. We compared the resulting BB source profile and the time-series of BB contributions to those of the original dataset without the additional tracers. The results reveal that differences are minor. The source profiles closely agree and there is a mean absolute difference of 3.5% (0-12.4%) in explained variances for the source components. The mean difference in the factor contributions to $PM_{2.5}$ for the winter of 2015-2016 was +0.68 $\mu g/m^3$ (higher in the original solution). The latter is analyzed in +0.59 $\mu g/m^3$ in day-time and +0.82 $\mu g/m^3$ in night-time average contributions. The mean difference in the fractional contribution to $PM_{2.5}$ is +3.4%.

The results regarding the uncertainty are summarized in the following figures and will be added as supplementary material.

[Figure]

Figure S2: Explained variances of PMF-components by the biomass-burning factor in the original solution (A) and in the solution including ACSM fragments for the winter 2015-2016 (B).

[Figure]

Figure S3: Comparison of calculated contributions ($\mu g \ m^{-3}$) of the biomass burning factors, in the original solution (A) and in the solution including ACSM fragments for the winter of 2015-2016 (B).

Below we list our response to the reviewer's specific comments (in italics) and the corresponding changes in the manuscript.

*Specific comments:*
*1. Page 2, line 24: It seems that there is something missing after "between winter 2012".*
We thank the reviewer for noticing and the missing word "winter 2013" was added.

*2. Page 2, line 12: Is there something missing after "the city center and" or should the "and" just be deleted?*
The reviewer is right the word "and" has been deleted.

*3. Page 3, lines 25-29, and further within the text: It seems that during winter separate day and night samples were collected in addition to the regular 24-h samples. However, later in the manuscript (e.g., in Figure 4(a) and Figure 5), it is not clear whether the winter data were derived from the regular 24-h samples or from the day and night samples.*

As mentioned in section 2.2 "During the three occurring winter periods (from December to February), two-month intensive campaigns (sampling frequency of 12h; n=447) were conducted". Thus, in winter mainly 12h samples were collected and the results at figures 4a and 5 reporting results during winter originate from these 12-h samples.

*4. Page 4, line 26: Abbreviations and acronyms (here PMF) should be defined (written full-out) when used for the first time within the body of the manuscript.*
*5. Page 7, line 23: Abbreviations and acronyms (here IM and SS) should be defined (written full-out) when used for the first time within the body of the manuscript.*

Most abbreviations and acronyms used throughout the manuscript have been defined when they were used for the first time within the body of the manuscript. For instance PMF/IM/SS have been defined in the abstract, Page 1, line 12. However, we thank the reviewer for helping us improve the manuscript and carefully checked for additional omissions.

*6. Page 8, lines 1-2: There are no sulphate and ammonium data in Figure 3.*

The reviewer is correct and this sentence has been corrected accordingly to represent exactly what the Figure illustrates.

*7. Page 8, line 8: It is unclear whether the number of samples (780) includes only 24-hsamples or also day and/or night samples.*

Since it was not clear, a comment has been included to specify that all samples both 12h and 24h have been considered.

*8. Page 8, line 15: It is unclear whether the winter PM2.5 mass was derived from the 24-h samples or from the day and night samples.*

As winter samples consist of 12h samples the $PM_{2.5}$ winter value derived from these samples. The reviewer is right and as it was not clear a comment has been included in the manuscript.

*9. Page 8, line 16, and further in the manuscript: Reference is made here to Figures 5a,b and later to subfigures for Figure 7, but the different subfigures of Figures 5 and 7 are not labeled with a, b, c, and so on.*

Indeed the reviewer is correct and figures have been labelled as suggested.

*10. Page 8, lines 22-23: There is no info on the aethalometer in Section 2. What size fraction was measured with it? And how was BCwb obtained? Some info and/or literature references on this are needed in Section 2.*

As instructed by the reviewer additional information are provided in the manuscript (Section 3.4.2), along with the appropriate reference.

*11. Page 9, line 16: The use of the word "between" is incorrect here. Could it be replaced by "of"?*
Done.

*12. Page 9, line 17: Why is the word "morning" used here? How were "morning" data obtained? Should it not be "day"?*
The word "morning" has been replaced by "day".

*13. Page 9, line 27: Why is there only one correlation coefficient given for the winter? Or were the same correlations obtained for OC and EC?*

Indeed, they both had the same correlation coefficient and a sentence was added in the manuscript on that direction.

*14. Page 10, lines 10-13: I do not understand the reasoning. If the molar ratio of NH4/nssSO4 is smaller than unity, would one not expect a mixture of H2SO4 and NH4HSO4?*

The reviewer is right and we meant equivalent ratio not molar ratio. It has been appropriately changed in the manuscript.

*15. Page 11, line 17: It is unclear what is meant by "increase by about 57%". Increase of what relative to what?*

As it is presented in table 2, $nssK^+$ presents a day to night increase equal to 54% once all winter samples are included, while equal to 57% if only SP events are included. That is the reason we refer to an "increase by about 57%". As requested by the reviewer an explanation was added.

*16. Page 13, lines 11-13: I cannot follow the reasoning. Would the correlations between the three elements during winter not suggest that wood burning is an important source, considering that this was also observed in the study of Maenhaut et al. (2016)?*

We fully agree with the reviewer and we do understand his/her point. Correlations of As, Cd and Pb during winter suggests emissions from wood burning as mentioned in Page 13, line 4-7. Furthermore, the fact that As and Cd, Pb have been associated with coal combustion further supports that hypothesis. The reference Maenhaut et al., 2016 was also added to indicate that other studies reach the same conclusion too.

*17. Page 16, line 9: Abbreviations and acronyms (here EF) should be defined (written full-out) when used for the first time within the manuscript.*

Done.

*18. Page 17, line 12: It is unclear what is meant by "fuel" here.*

It refers to fossil fuel use, and it was corrected accordingly in the manuscript.

*19. Pages 18-26, Reference list: There are several problems. For authors with more than 1 initial, there should be a space between the initials; each initial should be followed by a full stop (.); for references with at least 3 authors, there should be ", and" before the last author (note that for references with only 2 authors, there should not be ", and" but " and" instead; the publication year should be at the end of the reference and not after the authors; "Kawamura and Iskushima, 1993" (on page 21, lines 13-14) should be moved down to before "Kawamura et al., 1996"; "Uria-Tellaexte and Carslaw, 2014" (on page 25, lines 9-10) should be moved down to before "Valaoras et al., 1998".*

We thank the reviewer for raising these issues and we would like to apologise for our mistakes. All changes proposed have been performed

*20. Further problems with the references: The following references are in the text, but not in the Reference list: - page 14, lines 22 and 32: Amato et al., 2016.*
*- page 16, line 18: Dall'Osto et al., 2012; there is Dall'Osto et al., 2013 in the Reference list to which not is referred within the text.*

We thank the reviewer for helping us to improve the manuscript. All references have been checked.

*21. Page 31, caption of Figure 5: This caption should be extended. What does the line inside each box indicate? And what do the whiskers mean? Why do some whiskers end on a dash and other not?*
The reviewer is correct and in order to provide more information to the reader additional information have been included in the figure caption. The fact that some whiskers end on dash and other do not is due to the fact that in the latter we intentionally changed the maximum bound of the y axis in order to emphasize on the values presented.

*22. Technical and other (mostly minor) corrections:*
We would like to thank the reviewer for his/her time and all below mentioned technical and minor corrections have been considered.

*- page 2, line 2: replace "interest on" by "interest in".*
*- page 2, lines 3 and 20; page 8, line 24; page 11, line 32: replace "e.g." by "e.g.,".*
*- page 2, line 13, and on several occasions further in the manuscript: for references within parentheses there should be a comma before the publication year.*
*- page 2, line 30; page 4, line 2; page 7, line 3: replace "water soluble" by "watersoluble".*
*- page 2, line 33: replace "in the" by "in".*
*- page 3, line 8: replace "Athens," by "Athens".*
*- page 3, line 16; page 10, line 4: replace "long range" by "long-range".*
*- page 3, line 17: replace "Northern sector" by "northern sector".*
*- page 3, line 19: replace "Southern sector" by "southern sector".*
*- page 3, line 21: replace "Athens" by "Athens'".*
*- page 3, line 22: replace "are provided" by "is provided'".*
*- page 3, line 27: replace "to the" by "in the'".*
*- page 4, line 11: replace "species concentrations" by "species".*
*- page 4, line 13: replace "in ultrasonic bath and" by "in an ultrasonic bath and the extracts were" and replace "0.45_m" by "0.45 _m".*
*- page 4, line 17: replace "in details" by "in detail".*
*- page 4, line 20: replace "while" by "while for".*
*- page 4, line 27: replace "on 12-h" by "on the 12-h".*
*- page 4, line 31: replace "with analysis" by "with the analysis".*
*- page 5, line 10: replace "their contributions" by "the day and night contributions".*
*- page 5, lines 17-18: replace "probability (CBPF) function" by "probability function (CBPF)".*
*- page 5, line 29: replace "concentration of" by "concentrations of".*
*- page 6, line 8: replace "represents time" by "represents the time".*
*- page 6, line 22: replace "concentration of" by "concentrations of".*
*- page 7, line 14: replace "confined in" by "confined in the" and replace "they do not" by "do not".*
*- page 7, line 15: replace "exceed few" by "exceed a few".*
*- page 7, line 17: replace "trace element" by "the trace element".*
*- page 7, line 23: replace "Mass Closure" by "mass closure".*
*- page 7, line 27: replace "relative ratio" by "concentration".*
*- page 8, line 19: replace "low winds" by "low wind speeds".*
*- page 8, line 24: replace "in GAA" by "in the GAA".*
*- page 8, line 30: replace "i.e" by "i.e.,".*

*- page 8, line 32: replace "have been" by "has been".*
*- page 9, line 8: replace "winter POM" by "winter, POM".*
*- page 9, line 12: replace "on carbonaceous" by "on the carbonaceous".*
*- page 9, line 13: replace "Similar" by "A similar".*
*- page 9, line 19: replace "on fine" by "on the fine".*
*- page 9, line 26: replace "correlation with" by "correlation with the".*
*- page 9, line 28: replace "Composition" by "composition".*
*- page 10, line 1: replace "low winds" by "low wind speeds".*
*- page 10, line 21: replace "considerably reduce" by "are considerably reduced".*
*- page 10, line 22: replace "Similar" by "A similar".*
*- page 10, line 34: replace "concentrations in" by "in".*
*- page 11, line 6: replace "from Sahara" by "from the Sahara".*
*- page 11, line 7: replace "Regarding, nssK+" by "Regarding nssK+,".*
*- page 11, line 9: replace "from on line" by "from online".*
*- page 11, line 12: replace "present" by "exhibit a".*
*- page 11, line 17: replace "i.e." by "i.e.,".*
*- page 11, line 22: replace "exhibit" by "exhibits".*
*- page 11, line 23: delete ", respectively".*
*- page 11, line 25: replace "2011)" by "2011).".*
*- page 11, line 32: replace "e.g wood" by "e.g., wood".*
*- page 12, line 5: replace "as it is linked" by "and thus mainly linked".*
*- page 12, line 7: replace "maxima also" by "maxima are also".*
*- page 12, line 8: replace "road re-suspended" by "resuspended road".*
*- page 12, line 14: replace "Different" by "A different".*
*- page 12, line 16: replace "trends, corroborates" by "trends corroborate".*
*- page 12, line 18: replace "Anthropogenic" by "anthropogenic".*
*- page 12, line 19: replace "The analyzed" by "The measured".*
*- page 12, line 22: replace "they related" by "are related".*
*- page 12, line 26: replace "it is included" by "is included".*
*- page 12, line 29: replace "amongst other" by "and non-ferrous metals production amongst others".*
*- page 13, line 14: replace "in our site" by "at our site" and replace "significant" by "a significant".*
*- page 13, line 18: replace "seasonally basis" by "seasonal basis".*
*- page 13, line 28: replace "Factors" by "Factor".*
*- page 13, line 30: replace "PM2.5" by "The PM2.5".*
*- page 13, line 31: replace "n.s." by "not significantly different from 0".*
*- page 13, line 32: replace "indicating" by "indicated" and place "indicated the stability of the solution" at the end of the sentence on page 14, line 1.*
*- page 14, line 5: replace "Burning" by "burning".*
*- page 14, line 13: replace "or EC" by "of EC".*
*- page 14, line 16: replace "to 2.3" by "to the 2.3".*
*- page 14, line 21: replace "are abundant" by "is abundant".*
*- page 15, line 14: replace ", however" by "; however,".*
*- page 16, line 4: replace "Sources" by "sources".*
*- page 16, line 9: replace "to upper" by "to the upper".*
*- page 16, line 10: replace "8% of" by "8% of the".*
*- page 16, line 28: replace "to aerosol" by "to the aerosol".*
*- page 17, line 2: the parentheses around the percentages should be removed.*
*- page 17, line 4: replace "Ionic" by "The ionic".*

*- page 17, line 8: replace "significant" by "a significant".*
*- page 17, line 13: replace "lesser extend" by "lesser extent".*
*- page 17, line 19: replace "with higher" by "with a higher".*
*- page 17, line 21: replace "respectively" by "respectively,".*
*- page 17, line 31, and page 18, line 1: replace "acknowledges" by "acknowledge".*
*- page 23, line 7: replace "Nair, and" by "and Nair,".*
*- page 25, line 27: replace "Waenhaut" by "Maenhaut".*
*- page 30, line 1: replace "aerosol species" by "aerosol component or species".*
*- page 35, line 1: replace "to component" by "to the component".*
*- pages 36, 37 and 38: the Table headings should be above the Tables instead of below.*
*- page 36: the information on the Sampling, Location and Reference should be at the top of*
*Table 1 instead of at the bottom.*
*- page 37, line 3: replace "The annual" by "Annual".*
*- page 37, line 27: replace "elements during" by "elements and species during".*

*Comments for the Supplement:*
*Page 2: It is unclear what "CPF probability" indicates. In the Main text CBPF is mentioned.*
*Page 3, line 1: Replace "wind speed and speed" by "wind direction and speed in m s-1".*
*Page 4: The Table heading should be above the Table instead of below.*

Dall'Osto, M., Hellebust, S., Healy, R. M., Connor, I. P., Kourtchev, I., Sodeau, J. R., Ovadnevaite, J., Ceburnis, D., O'Dowd, C. D., and Wenger, J. C.: Apportionment of urban aerosol sources in Cork (Ireland) by synergistic measurement techniques, Sci. Total Environ., 493, 197-208, 2014.

Li, Z., Hopke, P. K., Husain, L., Qureshi, S., Dutkiewicz, V. A., Schwab, J. J., Drewnick, F., and Demerjian, K. L.: Sources of fine particle composition in New York city, Atmos. Environ., 38, 6521-6529, 2004.

---

## Author Comment (AC2) · 26 Jun 2018

*A long data series of PM2.5 speciation (more than 800 samples) obtained at an urban background site in Athens during a two-year period is presented. Description is focused on the winter period in order to quantify the impact of heating emissions and to define effective mitigation actions. A Positive Matrix Factorization was applied, identifying six sources for PM. The importance of domestic heating in winter is highlighted, and authors suggested a significant contribution of biomass burning (BB). However, this contribution can be overestimated (see below). Results obtained with an aethalometer are used for the interpretation of the BB contribution. However, these measurements are not properly described. The text is long and redundant. An effort must be made to reduce the length of the text. Some minor clarifications are needed before publication in ACP.*
We would like to thank the reviewer for his/her comments. We made an effort to decrease the length of the text, we removed figure 4 and deleted several details on the PMF methodology. Below we list our responses to his/her comments (in italics) and the corresponding changes in the manuscript.

*Page 3, section 2.2. Please, clarify the analysis performed for the 24h and the 12h samples. Were the 24h and 12h filters collected simultaneously during winter? Did you analyze total K, Mg and Na by ICP? Only the soluble fraction of these elements is presented, However, the contribution of the insoluble fraction of these elements can be important during dust events.*
During winter only 12h samples were collected. As it was not clear enough, a change in the manuscript has been performed to clarify this. Total elements were also measured by ICP but we decided not to include them in an effort to keep the length of the manuscript to a minimum as requested by the reviewers.

*Section 2.3: did you consider both, 24h and 12h filters for the PMF? Why did you perform the PMF analysis for the winter samples only? A higher number of cases may favor the identification of factors.*
Emphasis was given as the title indicates in wintertime. As said before during wintertime mainly 12h samples have been collected leading to a sufficient number of samples to allow for a robust PMF analysis. The reviewer is right that it is possible for additional factors to be extracted with more data. However, we have run models for the combined winter-summer dataset and did not identify further viable factors of significant contribution – at least for this site. Moreover, inclusion of summertime PMF would further increase the length of the manuscript.

*Section 3.1 and 3.2. There are more recent papers on PM2.5 levels and composition and source apportionment in Europe. See Amato et al., Atmos. Chem. Phys., 16, 3289-3309, 2016*
We thank we reviewer for pointing it out and they have been included in the manuscript.

*Section 3.2.1. Please, use ranges from <DL (specify for each component) to the maximum value; avoid referring to 0 concentrations.*

As pointed out by the reviewer reference to 0 concentrations has been avoided and replaced by <DL.

*Page 7; line 12. These elements (Mn, Cr, Ni, V, As: : :) may also be associated to mineral dust. I would say ": : :of major anthropogenic origin: : :"*
The reviewer is correct and the appropriate change has been performed in the manuscript.

*Section 3.3; page 7, line 31. How did you estimate the sea salt fraction of these PM components? Please, add references.*
As requested by the reviewer, information and the appropriate references have been included in Section 3.3.

*Page 8, section 3.4.1. The combined effect of meteorology (less developed boundary layer, low wind speed: : :)and the increase of the domestic emissions result in a poor air quality in winter. Thus, page 8 – line 28-30; the increase of PM2.5 concentrations during the night in winter may be related to the lower development of the boundary layer during the night and not only to the increase of the emissions of domestic heating.*
Based on boundary layer measurements and model estimates above Athens (e.g. Kassomenos et al., 1995) reported a decrease of BL during night-time during all seasons and in fact, the day-night BL height difference is more pronounced during the summer. Thus, the night-time accumulation of the pollutants during winter compared to summer clearly highlights the impact of additional emission sources related to heating.

*More information on the aethalometer measurements and model is needed. The description of the BC measurements should be included in the section 3.4.2 on carbonaceous components. Does the daily evolution of BCwb differ from that of BCff?*
Information on the aethalometer measurements and the model used was added to the manuscript in the section 3.4.2 as requested by the reviewer. On the other hand the daily evolution of the two components BCwb and BCff is indeed quite different as it was already described in Fourtziou et al., 2017 and Gratsea et al., 2017 and these two references were added now to the manuscript. No additional info was added as i) it is out of the scope of the work and ii) to avoid further increasing in the length of the manuscript.

*Page 10. Lines 28-32. The insoluble fraction of Mg, K and Na is usually associated to the dust fraction.*
We agree with the reviewer but here we refer to the soluble part not to the insoluble one.

*Page 11, lines 10 to 15. Mineral K is usually associated to clay minerals more than to carbonates. Could you check the correlation between K+ and Al for dust events and for non-biomass periods?*
As correctly stated by the reviewer mineral K is indeed associated with Al, given its crustal origin. Here $K^+$ refers to the soluble fraction whereas Al to the total (soluble and insoluble fraction). Despite this $K^+$ correlate with both Al and nss-$Ca^{2+}$ during dust events and out of wood burning periods. However as regression with nss-$Ca^{2+}$ has a better coefficient (most probably because both refer to the soluble fraction) we prefer to use nss-$Ca^{2+}$ to derive the wood burning fraction of $K^+$.

*Page 11, line 25. Please, replace : : :2011)" by : : :2011)."*
Indeed, the full stop was missing. Thank you for noticing.

*Page 14. Section 3.5.2 Line10. Please, add a figure or a reference for BB vs BCwb correlation.*
As stated before the variability of BC and BCwb during winter-time was already discussed in Fourtziou et al., 2017 and Gratsea et al., 2017 and these two references were now added to the manuscript.

*Line 11. Please, specify the period for the ACSM measurements carried out by Bougiatioti et al., 2014. The PMF profile for BB is characterized by a high contribution of nitrates. This correlation can be due to covariation induced by meteorological factors Authors explained the association of nitrates with the BB factor by high emission of NOx by biomass burning and by the lower acidity. However, as explained later NOx better correlated with the vehicular emission factor. May a fraction of nitrates be derived from fuel oil emissions? Then, the contribution of BB may be overestimated.*
First, we clarify that the reference to the work of 2014 was included to indicate the ACSM methodology and referred to measurements performed in Crete. We have updated the text to mention the ACSM measurements in Athens which cover the full period 2016-2017 as well as the winters in 2014-2015 and 2015-2016 and are presented in Stavroulas et al. (2018).
We agree with the reviewer pointing the relevance of meteorological conditions for the observed variability, especially for the present analysis which is conducted in a sub-daily basis. As already stated in the manuscript, this variability is of importance for the atmospheric chemistry processes. The presence of $NO_3^-$ in the BB factor can be explained by increased night-time local emissions of $NO_x$ from biomass burning and by the lower acidity occurring during BB events, which as shown in Bougiatioti et al. (2014) favours the partitioning of $NO_3^-$ in the particulate phase. It should be mentioned that except $NO_x$, biomass burning is also a significant local source of ammonia emissions (Zhou et al., 2015), which otherwise in the area of Athens are very limited (Fameli and Assimakopoulos, 2016). These are important factors for rapid ammonium nitrate formation especially at low temperatures (Paulot et al., 2017) encountered in our site during the night in winter.
We have clarified in the text that $NO_x$ concentrations were used as measured at a nearby kerbside traffic site, merely as a proxy of traffic variability in the area (Grivas et al., 2018), in order to validate the vehicular factor. At the urban-background site location, direct emissions from road fuel combustion are of minor importance.
Numerous efforts to extract a separate secondary nitrate factor were conducted, however this was not possible, while in every case nitrate was principally associated with the BB factor. Solutions with more than six factors also did not perform adequately in terms of rotational uncertainty, having an increased number of incorrectly assigned factors and numerous swaps in the BS-DISP procedure.
According also to the suggestion of Reviewer #1, the 6-factor PMF solution was validated using additional biomass-burning tracers. Since measurements of levoglucosan were not performed in this work, PMF was again run adding data for the *m/z* 60 and *m/z* 73 fragments from ACSM measurements concurrently conducted at the site during the winter of 2015-2016. At their study performed during wintertime in Athens using various wood burning tracers, Fourtziou et al. (2017) have reported significant correlations between, nss-$K^+$, levoglucosan, and ACSM-derived

*m/z* 60, the last being considered as a fingerprint of levoglucosan. The results of the repeat analysis did not differ significantly from those reported in the original manuscript and $NO_3^-$ was again associated with the BB source.

Our experience and other studies with source apportionment for $PM_{2.5}$ in Athens (Paraskevopoulou et al., 2015; Diapouli et al., 2017) indicate that an independent nitrate factor is difficult to isolate, due to the widespread presence of direct sources in the area, the fast chemical conversion from emissions and the dependence of its partitioning on atmospheric conditions.

The chance that an overestimation could be due to nitrates is now mentioned in the revised manuscript, according to the reviewer's suggestion. We believe that the BS-DISP error margins of Figure 8, can provide an indication of the uncertainty related to nitrate determination. As an upper limit of overestimation due to nitrate, we considered the total nitrate assigned to the BB factor (as $NH_4NO_3$). In this case, the overestimation during day-time and night-time would be 0.58 and 1.18 μg m$^{-3}$, respectively. The respective contributions of the factor to $PM_{2.5}$ would be 16% and 35% during the two periods of the day (3% and 4% less than the reported solution, respectively). For comparison, Amato et al. (2016) have reported an annual secondary nitrate contribution of 0.7 μg m$^{-3}$ for $PM_{2.5}$ in suburban Athens. The overall contribution of the factor to $PM_{2.5}$ mass would be then 28% (from 32%), which is still a considerably large value. Indicatively, Diapouli et al. (2017) have reported annual mean contributions of BB at sites in Athens in the range of 23-46% of $PM_{2.5}$.

*Page 15. The vehicular source is characterized by a relatively higher contribution during night in the winter period. Can be this diurnal trend related to a significant contribution of fuel oil domestic heating? The vehicular source highly correlated with BCff; does the diurnal trend of BCff reproduce the trend of vehicular traffic? Is there other source of BCff during night in winter?*

It is true that some involvement from heating fuel combustion in the vehicular emission factor can't be overruled but it is very difficult to account for, quantatively. Especially in Greece, due to large tax hikes in heating oil, it has been reported that occasionally road transport diesel is used instead for residential heating. There have also been issues with road diesel fuel adulteration by heating fuel oil (Kalligeros et al., 2003).

However, the relative contribution of space heating from oil to aerosol emissions in central Athens, has always been considered of secondary importance (Economopoulos, 1997). Even more so, in the years of the recession in Greece, the use of oil for space heating has reached a low-point. Fameli and Assimakopoulos (2016) have recently estimated that the share of heating oil is less than 2% in the total $PM_{10}$ emissions from domestic heating in Greece, the rest emanating from fire-places and wood stoves.

The diurnal variation of road traffic in Athens (Grivas et al., 2012; Fameli and Assimakopoulos, 2015) can reproduce the variability of BCff at the Thissio site (Fourtziou et al., 2017), if the diurnal variability of meteorological factors is taken into account (BL height changes and increased wind speeds in the afternoon). The typically bimodal circadian variation of BCff generally coincides with those of traffic-related gaseous pollutants and of $PM_{10}$ at traffic stations in Athens (Grivas et al., 2008).

It is noted that, in the center of Athens, where the site is located, due to increased commercial and recreational activity, late-afternoon and evening traffic is not negligible. In addition, there are traffic restrictions for passenger vehicles in the center of Athens during daytime (7:00-20:00, based on odd-even plates system). Data from traffic sensors on roads inside the restricted

circulation zone, during December-January, show that night-time/daytime traffic volumes are comparable (ratio of 0.96). This, combined to the formation of the stable nighttime boundary layer at low heights can lead to an increase of concentrations.

Overall, as stated in the manuscript, the night-time contributions of the vehicular source is slightly higher (by a factor of 1.5) compared to day-time, a difference not assessed as statistically significant at the 0.05 level, while the contribution of this source to $PM_{2.5}$ is similar during night and day (19%). However, since a minor influence of domestic heating can't be excluded, we specifically mention it in the revised text.

*Page 15, Line 20. V/Ni derived from fuel oil combustion sources is usually associated to SO42-; however, SO42- is not significantly associated in this factor*

We agree with the reviewer that in some cases stronger associations between fuel-oil combustion and sulfate have been reported. However, we note that sulfate is present in the oil combustion source profile (we have selected a different marker in the corresponding Figure to make this clearer).

Due to the identification of additional sources closely associated with sulfate and to the small overall contribution of the oil combustion factor to $PM_{2.5}$ mass, its explained variance by the factor is relatively small (4%). This value is in line with Amato et al. (2016) which have attributed fine particulate S to fuel oil combustion at rates of 2%-13%, at a suburban background site in Athens and at an urban background site in Barcelona, respectively. Also, Reche et al. (2012), have displayed an explained variability circa 5% for fine particulate S in the fuel oil combustion factor for Barcelona. Pandolfi et al. (2011), at the heavy-oil emission hotspot of Gibraltar, have reported that the explained variance of the relevant factor for sulfate that is around 10%.

The mass fraction of sulphate in the source profile (0.07μg/μg) is comparable to results from Grivas et al. (2018) that calculated for the fuel oil combustion profile in $PM_{2.5}$ values of 0.08 and 0.15 μg/μg at urban background and traffic sites in the center of Athens, respectively. Additionally, the source profile reported by Kocak et al. (2011) in Istanbul contains sulfate at approximately 0.05 μg/μg. Therefore, we considered the sulfate association in the oil-combustion factor to be within the range reported by past studies in Athens and other cities in the Mediterranean.

References

Economopoulos, A. P.: Management of space heating emissions for effective abatement of urban smoke and $SO_2$ pollution, Atmos. Environ., 31, 1327-1337.

Fameli, K. M. and Assimakopoulos, V.D.: Development of a road transport emission inventory for Greece and the Greater Athens Area: Effects of important parameters, Sci. Total Environ. 505, 770-786, 2015.

Fameli, K. M. and Assimakopoulos, V. D.: The new open Flexible Emission Inventory for Greece and the Greater Athens Area (FEI-GREGAA): Account of pollutant sources and their importance from 2006 to 2012, Atmos. Environ., 137, 17-37, 2016.

Grivas, G., Chaloulakou, A., and Kassomenos, P.: An overview of the particle pollution problem in the Metropolitan Area of Athens, Greece. Assessment of controlling factors and potential impact of long range transport, Sci. Total Environ., 389, 165–177, 2008.

Kalligeros, S., Zannikos, F., Stournas, S., and Lois, E.: Fuel adulteration issues in Greece, Energy, 28, 15-26, 2003.

Kassomenos, P., Kotroni, V., and Kallos, G.: Analysis of climatological 710 and air quality observations from Greater Athens Area, Atmos. Environ., 29, 3671-3688, 1995.

Koçak, M., Theodosi, C., Zarmpas, P., Im, U., Bougiatioti, A., Yenigun, O., and Mihalopoulos, N.: Particulate matter ($PM_{10}$) in Istanbul: Origin, source areas and potential impact on surrounding regions, Atmos. Environ., 45, 6891-6900, 2011.

Pandolfi, M., Gonzalez-Castanedo, Y., Alastuey, A., de la Rosa, J. D., Mantilla, E., Sanchez de la Campa, A., Querol, X., Pey, J., Amato, F., and Moreno, T.: Source apportionment of $PM_{10}$ and $PM_{2.5}$ at multiple sites in the strait of Gibraltar by PMF: impact of shipping emissions, Environ. Sci. Pollut. Res., 18, 260–269, 2011.

Paulot, F., Paynter, D., Ginoux, P., Naik, V., Whitburn, S., Van Damme, M., Clarisse, L., Coheur, P. F., and Horowitz, L. W.: Gas-aerosol partitioning of ammonia in biomass burning plumes: Implications for the interpretation of spaceborne observations of ammonia and the radiative forcing of ammonium nitrate, Geophys. Res. Lett., 44, 8084-8093, 2017.

Reche, C., Viana, M., Amato, F., Alastuey, A., Moreno, T., Hillamo, R., Teinilä, K., Saarnio, K., Seco, R., Peñuelas, J., Mohr, C., Prévôt, A. S. H., and Querol, X.: Biomass burning contributions to urban aerosols in a coastal Mediterranean City, Sci. Total Environ. 427-428, 175-190, 2012.

Stavroulas, I., Bougiatioti, A., Paraskevopoulou, D., Grivas, G., Liakakou, E., Gerasopoulos, E., and Mihalopoulos, N.: Sources and processes that control the submicron organic aerosol in an urban Mediterranean environment (Athens) using high temporal resolution chemical composition measurements, Atmos. Chem. Phys. Discuss., https://doi.org/10.5194/acp-2018-356, in review, 2018.

Zhou, Y., Cheng, S., Lang, J., Chen, D., Zhao, B., Liu, C, Xu, R., and Li, T.: A comprehensive ammonia emission inventory with high-resolution and its evaluation in the Beijing-Tianjin-Hebei (BTH) region, China, Atmos. Environ., 106, 305-317, 2015.

---

## Author Comment (AC3) · 26 Jun 2018

*The paper by Theodosi et al. reports about the large observational study monitoring particulate matter sources in central Athens emphasizing contribution of biomass burning, especially during winter and in particular during the night. The study presents no new knowledge, nor scientific advancement, although it provides very detail account of sources and is methodologically correct. The paper is often buried in details and often repetitive and better focus on the main idea is necessary. At the moment the paper looks more like a report than scientific paper. Better focus and summary at the end of sections is necessary and perhaps separating results from discussion. The paper ends abruptly with conclusions as an extensive summary. Quite an improvement (including the English) is needed before the paper can be accepted for publication.*

We would like to thank the reviewer for his/her comments. We disagree that no new knowledge is provided. Similar works on long-term aerosol chemical characterization are very scarce in Europe. Especially in Greece, this is the first time that daily and 12-h filter samples have been analysed for ions, OC/EC and trace metals to provide a complete chemical characterisation in the largest urban centre for more than two years. Although not easy, as the reviewers have asked for additional details, we tried to decrease the length of the manuscript by removing several sentences and one figure to the supplementary material. Below we list our detailed response to his/her comments (in italics) and the corresponding changes in the manuscript.

*Comments:*
*I guess the authors are overstretching when repeatedly refer to the Greater Athens Area, because a single station in the city centre can hardly described the entire Greater Athens area. Clearly more stations are necessary to monitor greater agglomerate area due to different population/build up density, anthropogenic activities and traffic. Therefore, a better reasoning is required to refer to GAA if possible at all. Very often the authors go into unnecessary simplification and repetition, e.g. regarding temperature inversions and limited mixing. Every specialist is aware of the phenomenon and mentioning it once in detail is enough.*
It is clear that several stations are required to obtain a complete picture for a large agglomeration such as GAA but on the other hand it is understandable that it would be very difficult to perform such a chemical analysis in-depth for several stations. As a reminder, in the present work more than 800 filters have been collected and analysed for main ions, OC/EC and metals with emphasis (twice daily) on the winter period. As shown in Gratsea et al. (2017), by comparing Thissio data with several (5) stations of the National Air Pollution Monitoring Network, our station captures quite well the temporal variability of primary pollutants like CO within the GAA. In addition Thissio station is not directly impacted by sources and its position on a hill at the center of Athens allows it to be considered representative of the background in the wider area. For that reason we consider the station as representative as possible for the GAA, at least for its central part. It is also worth noticing that during the sampling period, this was the sole urban background station monitoring PM concentrations in the densely-populated central Athens (gathering 1.03 million people, almost

a third of the GAA population), and its results can be considered relevant for the exposure of a large part of GAA inhabitants.

"*The high correlation between the five NAPMN sites and Thissio, as well as the relative difference in absolute levels, support the characterisation of Thissio as an urban background site, non-intensively affected by local traffic*; From Gratsea et al., 2017).
However as suggested by the reviewer the term GAA is now not mentioned in the manuscript. Also atmospheric conditions are now mentioned only once in the manuscript.

*Minor comments:*
*Page 2, line 6. through emission reduction measures.*
*Line 9. : : :toxic and carcinogenic components.*
*Line 12. Trace metals are also related to chronic and acute...*
*Line 18. : : :such as traffic and industrial activities.*
*Line 19. since the winter 2011-2012.*
*Line 21. the great impact.*
*Line 25. Linking them to the presence*
*Line 33. : : :has been undertaken in Southern Europe, offering challenging conditions (what is challenging there by the way?)*
*Page 3, line 2. the current work was focused on winter period.*
*Line 4. The aerosol sources during the night are not new. Rephrase to "... highlight the impact of night time PM sources".*
*Line 10. It is still representative of central Athens, especially during high pollution events during stagnant meteorological conditions. A single site cannot represent the whole GAA as suburban areas were not monitored. It is methodologically wrong to assume that a single site can serve as a reliable average of the entire GAA.*
*Line 13. why this word of caution when the next sentence explains everything?*
*Line 17. On a yearly basis, air masses of Northern origin from central and eastern Europe account for almost two thirds of the time.*
*Line 20. Ventilation is a poor term. You should be stating about stagnant conditions during which most severe pollution events have been occurred persisted for X % of time.*
*Line 28. Unclear. Were the 447 samples in addition to 848 samples or a fraction of them?*
*Line 30. Controlled RH=%?? conditions. Please specify.*
*Page 4, line 17. : : :in detail by Theodosi: : :*
*Line 26. Not contrast, but patterns, as no one can know in advance if they will contrast.*
*Line 27. trivial repetitive sentence.*
*Page 5, line 9. unclear - combined OR night-day?*
*Line 27. Was it really lower limit at 0.3ug/m3 which is inconceivably low for urban PM2.5 concentration. Was it realistic and how did it compare to chemical mass?*
*Page 6, line 8. zero OC or EC cannot be observed anywhere on Earth (even in Antarctica there is observable BC of _1ng/m3). Refer to below DL of Xug/m3.*
*Line 10. To be mathematically correct one should only present the arithmetic average if data are normally distributed in which case arithmetic average and median are closely similar number. We don't see that therefore data are lognormally distributed in which case the range and the median should be presented only. It has to be consistently presented either median alone or ALWAYS with the median, rather then selectively presenting median.*
All the aforementioned suggestions have been performed.
*Line 26. Excellent agreement should not be necessarily expected, "results were in close agreement". Close agreement suggests long-range transport as the main source.*

*Page 7, line 17. remained within the same order of magnitude between studies with only slight differences in between. Chemical mass closure. Given that all major and minor particulate matter species were measured, the chemical mass closure has been attempted. However, given the large uncertainty in OM/OC ratio and no data for the current study, how useful the chemical closure exercise was?*

An OM/OC ratio equal to 1.8 determined from Aerosol Chemical Speciation Monitor (ACSM) measurements at the same location, within the study period (Stavroulas et al., 2018) was used to perform a more accurate mass closure.

*Page 8, lin 19. Could further limit dispersion of pollutants.*
*Line 31. 289 days with smog conditions ...*
*Page 9, line 29. No statistically significant seasonality can be seen in SO4 time series considering error bars. Note the lowest concentrations in November-December 2014. Rewrite as the text is not reflecting the Figure.*

The text was rewritten as " $SO_4^{2-}$ concentration in the $PM_{2.5}$ fraction presented no statistical significant seasonality with the lowest values in winter"

*Page 10, line 4. What is the anthropogenic source of SO4 during summer if long-range transport is blamed? Is it not biogenic (Mediterranean DMS) instead?*
*Page 11, line 3. : : :suggesting insignificant influence from wood burning. Is it impossible that fraction of Na increase is due to higher primary in winter?*

Yes it could be possible but as both Cl and Mg are decreasing during night and Na is not, this could be an indication of a small contribution from heating and especially biomass burning, as previously reported by Fourtziou et al. (2017).
*Page 12. 3.4.4a Crustal elements*
*Line 26. Present in tailpipe emissions, not included.*
*Page 14. Lines 31-34. Reword – confusion with mean annual contributions and "such large wintertime contributions" – Corrected.*
*Page 15, line 13. What is the reason for heavy duty vehicles in central Athens?*

Karageorgos and Rapsomanikis (2010) have also mentioned the presence of heavy vehicles in central Athens, as a factor for particle emissions. The main reason is bus circulation, most probably of tourist buses, whose traffic is relatively dense in the area of the site, where various attractions are located.
*Line 23. westerly advections, not flows* - Corrected.
*Line 28. there is no information whether particles were internally or externally mixed. And even if known there is no sulphate particle in existence* - Corrected.

We would like to thank the reviewer for his/her time and all below mentioned technical and minor corrections have been considered.

---

## Author Comment (AC4) · 26 Jun 2018

We would like to thank the editor for his comment which will be taken into consideration in the revised version and will be adequately referenced. Note however that the work on ACSM addresses the sources and origin of organic submicron particulate matter in Athens during summer and winter. Thus there is no overlap in the objectives and methodology and partly address the issue of wood burning by using another approach.

---

## Author Response (AR3)

Dear Editor,

we submit the revised manuscript following the requests of reviewer #2. We hope that we have addressed all issues in a satisfactory way. Thank you for your consideration of the manuscript.

**Response to Reviewer #2**

We would like to thank the reviewer for his suggestions that helped improve the manuscript substantially. We list below a detailed response to hers/his comments.

*In a previous review I suggested applying the PMF for the complete database, considering the whole sampling period. As stated in the answers to reviewers, the authors "have run models for the combined winter-summer dataset and did not identify further viable factors of significant contribution". I still consider that a PMF covering the whole period will provide valuable information on sources and composition of PM2.5 in the study area. I would suggest to indicate in the text that this exercise was done and to include the results obtained in the supplementary.*

According to the suggestion of the reviewer, indicative PMF results using daily data from two full years (March 2014-February 2016) are now presented in the supplement. The PMF methodology regarding selection of factors and model validation is the same as the winter-time analysis. Supplementary figures display the percentage of species attributed to the six factors (S2) and their mean contribution to the $PM_{2.5}$ mass (S3). A decrease of the BB factor contribution in regard to winter is obvious (down to 13%, 2.3 μg $m^{-3}$). This implies that BB emissions during the non-winter periods, while still present, are much less intense and more of sporadic nature, associated to regional transport of agricultural burning and forest-fires (Sciare et al., 2008; Bougiatioti et al., 2014). The estimated mean annual fractional contribution of 13% is elevated compared to those reported by Grivas et al. (2018) at an urban background location in central Athens (8%) during 2011-2012 and by Amato et al. (2016) at a suburban background site (10%) during 2013. The juxtaposition of these results obtained for consecutive years within the recession period, at background sites in Athens, possibly underlines the establishment and evolution of the biomass burning issue.

In addition, on a year-round basis, the prominence of the secondary aerosol factor at the urban background scale emerges (reaching 38%), in consistence with past findings reported for the area and other European cities.

These two key differences from winter results are now discussed in the revised manuscript.

*For some elements, authors did analyze the whole fraction (soluble and insoluble, ICP-OES analysis form the acidic digestion) and the soluble fraction. However they only used the soluble fraction for the interpretation. In the previous revision I suggested to use also the insoluble fraction for interpretation but authors decide not to include it for no increasing the length of the text. I'd like to highlight that the average concentration of the insoluble Ca was 585 ng/m3 (as deduced form Table1) being the fifth*

*contributor to PM2.5 (after OC, EC, sulfate and ammonium). Including both the insoluble and the soluble fractions of Ca in the PMF would probably improve the identification of sources.*

We agree with the reviewer that both water-soluble and insoluble fractions have been occasionally utilized in PMF analysis (Rizzo and Scheff, 2007; Chalbot et al., 2013), although in the majority of cases this is

5  avoided in order to exclude bias related to double-counting of species (Hasheminassab et al., 2014). In view of this and as per the reviewer's suggestion, we have incorporated the water insoluble fraction of Ca (from now on and in the manuscript referred as Ca-ins) in our analysis, by subtracting ionic $Ca^{2+}$ from the total Ca quantified by ICP-OES following acid digestion (Beuck et al., 2011; Yubero et al., 2011). The uncertainty of the new variable was calculated following standard rules for propagation of

10  uncertainty.

Having a signal-to-noise ratio of 0.8 and according to the typical variable classification scheme followed so far in the study, Ca-ins was included as down-weighted species with increased uncertainty. Moreover, we would like to mention that the study average concentrations of soluble and insoluble Ca, displayed on Table 1 and mentioned by the reviewer, are higher than those recorded during the winter periods of the

15  PMF study, due to the temporal variability already discussed in the text. In fact, during the two-winter periods included in the PMF analysis, the average concentrations of $Ca^{2+}$ and Ca-ins were 51 and 270 ng $m^{-3}$, respectively.

As a result, the outcome of the PMF analysis including the Ca-ins fraction, indicated an improvement of source characterization, although not very pronounced. Differences of source contributions did not exceed

20  0.2 $\mu g\,m^{-3}$ or 0.8%. The solution explained 1% more of the PM2.5 mass, limiting the unaccounted fraction to 5%. Moreover, in the new run, slightly increased correlations with their respective external tracers were achieved for the vehicular and biomass burning factors, leading to the overall better differentiation among the sources.

Regarding the apportionment of the Ca-ins mass, it was split between the soil, vehicular and

25  secondary/regional factors (Yuan et al., 2008; Hellack et al., 2015; Huang et al., 2017). The highest fraction was attributed to the dust source, reflecting the impact of natural dust resuspension in the calcite-rich area of the central Athens basin (Argyraki and Kelepertzis, 2014). The insoluble Ca in Athens is expected mainly in the form of calcium carbonate (Sillanpaa et al., 2005). Overall, the obtained ratio of Ca (soluble+insoluble)/Al of 0.9 is comparable to that recently reported for PM2.5 at an urban background

30  site in central Athens (Grivas et al., 2018). Insoluble Ca associated with the vehicular factor is most probably due to traffic-induced dust resuspension (Kassomenos et al., 2012) and also due to its use as an additive in lubricants (Lough et al., 2005). Finally, we note the characteristic absence of the component in the sea salt factor, in contrary to $Ca^{2+}$ (Tan et al., 2016).

The findings of the analysis with Ca-ins have been integrated in the revised text, and all relevant values,

35  tables and figures have been updated accordingly.

*Did the authors evidence a time variation in the contribution of the biomass source to PM2.5 along the study period comparing the different winter periods? Do these results agree with those obtained from the aethalometers? Given the longer period covered by this paper, this can be a novel input compared with the other studies in the same topic in the area.*

We thank the reviewer for this insightful suggestion. We have compared mean contributions of the BB source and aethalometer $BC_{wb}$ concentrations, between the two winter periods of 2014-15 and 2015-16. The results revealed a decrease in both during the second period, with the average contribution of BB during winter 2015-16 being 33% lower than in winter 2014-15, while the respective decrease of $BC_{wb}$ was 41%. Both differences were statistically significant at the 0.01 level.

Meteorological parameters also varied between the two periods. Cold conditions were relatively harsher during the first winter, with the mean minimum daily temperature being lower by 1.5°C and more frequent northern synoptic-scale winds transporting cold air masses, factors amplifying the total demand for heating and leading to increased BB emissions.

Moreover, according to data from the Hellenic Statistical Authority, the amount of heating oil sold in Greece in 2015 increased by 51% compared to the previous year, reaching 0.41 mil metric tons (as opposed to 0.9 mil before the recession). It appears that a gradual stabilization of the biomass burning trend was in progress in Greece, as more residents slowly reverted back to the previously used heating fuel, aided by the rationalization of prices (e.g. up to 20% between 2014 and 2015, mostly due to declining crude oil spot prices). Thus, the use of biomass burning for space heating during the second winter is reasonably expected to have decreased. However, it is necessary to track this trend further in time, since heating oil prices have again risen during the last two years and uncertainty surrounds the future of the imposed excise tax on heating oil, which has originally led to the intensification of wood-burning practices in Athens.

Minor Comments

*I suggest modifying the title by: the "multiyear chemical composition of PM2.5 in Athens, with emphasis on the contribution of residential heating in winter time". I believe this is more appropriate.*
The title has been modified.

*Page 5, Line 22. Please indicate the period corresponding to the concentration of sulfate.*
The study period is now mentioned in the text.

*Page 9, lines 6-7. Higher levels of nitrate during night in the SP periods are not enough to conclude a major origin of nitrate related to heating.*
The phrase has been amended in order to imply additional factors for nitrate enhancement during nighttime.

*Page 12, line 12. The paper by Kalogridis et al. 2018 determined a contribution of BB to eBC of around 30% for the winter 2014-2015. You may include this reference here.*
The results of Kalogridis et al. (2018) are brought up in the revised text.

*References (not included in original manuscript)*

Beuck, H., Quass, U., Klemm, O., and Kuhlbusch, T.A.J.: Assessment of sea salt and mineral dust contributions to $PM_{10}$ in NW Germany using tracer models and positive matrix factorization, Atmos. Environ., 45, 5813-5821, 2011.

5    Chalbot, M.-C., McElroy, B., and Kavouras, I.G.: Sources, trends and regional impacts of fine particulate matter in southern Mississippi valley: Significance of emissions from sources in the Gulf of Mexico coast, Atmos. Chem. Phys., 13, 3721-3732, 2013.

Hasheminassab, S., Daher, N., Saffari, A., Wang, D., Ostro, B. D., and Sioutas, C.: Spatial and temporal variability of sources of ambient fine particulate matter ($PM_{2.5}$) in California, Atmos. Chem. Phys., 14, 12085-12097, 2014.

10   Huang, X., Liu, Z., Liu, J., Hu, B., Wen, T., Tang, G., Zhang, J., Wu, F., Ji, D., Wang, L., and Wang, Y.: Chemical characterization and source identification of $PM_{2.5}$ at multiple sites in the Beijing–Tianjin–Hebei region, China, Atmos. Chem. Phys., 17, 12941-12962, 2017.

Kassomenos, P., Vardoulakis, S., Chaloulakou, A., Grivas, G., Borge, R., and Lumbreras, J.: Levels, sources and seasonality of coarse particles ($PM_{10}$-$PM_{2.5}$) in three European capitals - Implications for particulate pollution control, Atmos. Environ.,

15   54, 337-347, 2012.

Lough, G.C., Schauer, J.J., Park, J.-S., Shafer, M.M., Deminter, J.T., and Weinstein, J.P.: Emissions of metals associated with motor vehicle roadways, Environ. Sci. Technol., 39, 826-836, 2005.

Rizzo, M.J., and Scheff, P.A.: Fine particulate source apportionment using data from the USEPA speciation trends network in Chicago, Illinois: Comparison of two source apportionment models, Atmos. Environ., 41, 6276-6288, 2007.

20   Sciare, J., Oikonomou, K., Favez, O., Liakakou, E., Markaki, Z., Cachier, H., and Mihalopoulos, N.:    Long-term measurements of carbonaceous aerosols in the Eastern Mediterranean: Evidence of long-range transport of biomass burning, Atmos. Chem. Phys., 8, 5551-5563, 2008.

Sillanpää, M., Frey, A., Hillamo, R., Pennanen, A.S., and Salonen, R.O.: Organic, elemental and inorganic carbon in particulate matter of six urban environments in Europe, Atmos. Chem. Phys., 5, 2869-2879, 2005.

25   Tan, J., Duan, J., Zhen, N., He, K., and Hao, J. Chemical characteristics and source of size-fractionated atmospheric particle in haze episode in Beijing, Atmos. Res.,167, 24-33, 2016.

Yuan, H., Zhuang, G., Li, J., Wang, Z., and Li, J. Mixing of mineral with pollution aerosols in dust season in Beijing: Revealed by source apportionment study, Atmos. Environ., 42, 2141-2157, 2008.

[revised manuscript text omitted]